# Efficient Training of Minimal and Maximal Low-Rank Recurrent Neural Networks

**Anushri Arora**
Department of Computer Science
Princeton University
aa1698@princeton.edu

**Jonathan W. Pillow**
Princeton Neuroscience Institute
Princeton University
pillow@princeton.edu

## Abstract

Low-rank recurrent neural networks (RNNs) provide a powerful framework for characterizing how neural systems solve complex cognitive tasks. However, fitting and interpreting these networks remains an important open problem. In this paper, we develop new methods for efficiently fitting low-rank RNNs in "teacher-training" settings. In particular, we build upon the neural engineering framework (NEF), in which RNNs are viewed as approximating an ordinary differential equation (ODE) of interest using a set of random nonlinear basis functions. This view provides geometric insight into how the choice of neural nonlinearity (e.g. tanh, ReLU) and the distribution of model parameters affects an RNN's representational capacity. We show that this perspective leads to an online training method that achieves higher accuracy with smaller networks than previous methods such as FORCE, and outperform backprop-trained networks of similar size while requiring substantially less training time. We then consider the problem of finding minimal and maximal low-RNNs for approximating a target dynamical system. We show that a variant of orthogonal matching pursuit (OMP) can be used to find the smallest RNN for a dynamical system of interest. At the other extreme, a dual space formulation allows for efficient fitting of infinite low-rank RNNs, which provide a Gaussian Process (GP) prior over dynamical systems. We use the resulting GP marginal likelihood to optimize the hyperparameters governing neural activation functions, which leads to improved training performance even for finite RNNs. Finally, we describe active learning methods for low-rank RNNs, which speed up training through the selection of maximally informative activity patterns.

## 1 Introduction

Recurrent neural networks (RNNs) are a popular tool for characterizing the computational properties of neural populations and the dynamics underlying complex cognitive tasks [1–6]. Previous work has proposed a variety of methods for training RNNs to implement a dynamical system or generate a target signal of interest, including reservoir computing [7, 8], FORCE [9, 10], and back-propagation [11–13]. However, trained RNNs remain difficult to interpret. Common approaches tend to use search methods to identify fixed points or "slow points", and then use dimensionality-reduction methods to visualize projected flow fields around these points [12, 14]. However, fixed-point finding algorithms are difficult to apply to high-dimensional systems, and it is often unclear how accurately low-D projections reflect a network's true dynamics [2, 15].

To overcome these difficulties, recent work has focused on "low-rank RNNs", in which the recurrent weight matrix is constrained to have low rank [3, 16, 6, 17]. This literature has shown that a wide variety of tasks can be implemented in low-rank RNNs, which exhibit dynamics whose dimension is limited by the rank of the recurrent weight matrix and number of input dimensions.

39th Conference on Neural Information Processing Systems (NeurIPS 2025).

A parallel arm of research has focused on methods for embedding low-dimensional quantities into high-dimensional network activity [18–23]. Of particular relevance to our work is the Neural Engineering Framework (NEF) which provides an analytical method for determining connection weights between neurons that implement a desired function or dynamical system. This is achieved via distributed representations, where each neuron characterizes the target function through a random nonlinear projection, forming diverse tuning curves across the population. Network connection weights are then obtained by finding the optimal linear combination of these nonlinear projections via least-squares regression, enabling complex function approximation.

This approach is closely related to sparse linear approximation methods that seek to use an overcomplete library of basis functions to fit the ODEs [24, 25]. For instance, SINDY proposes using sparse regression over a pre-specified polynomial basis to reconstruct a system's dynamics [26]. Moving-window approaches [27] employ a sliding-horizon scheme: solving a constrained optimization at each time window, and using statistical tests to prune irrelevant basis functions, thereby maintaining parsimony over time. More recently, [28] considered linear splines as basis functions, preserving network expressivity while yielding analytically tractable update rules that improve interpretability. However, these methods rely on explicit parametric representations of the system dynamics. In contrast, neural-network Gaussian processes (NNGPs) offer a nonparametric alternative, encoding priors over function classes defined by infinite-width neural networks [29–31]. While this has traditionally been limited to feedforward networks, recent work has extended it to recurrent settings via untied-weight constructions [32–34].

Here we present a unified framework for designing and training low-rank RNNs that implement a target dynamical system of interest. We begin with an offline method based on NEF, where each neuron acts as a nonlinear basis function and least-squares regression embeds a known ODE into the network. This yields a geometric view of universal approximation, illustrating how nonlinearity choices ($tanh$ vs ReLU) shape representational capacity. We then extend this to an online setting using a recursive least-squares update, and show that our method outperforms FORCE and BP in network accuracy and convergence speed. Next, we address the critical question of basis selection: using a variant of orthogonal matching pursuit (OMP), we identify the minimal set of basis functions needed to implement a given dynamics. We then consider infinite low-rank RNNs, which converge to a Gaussian process (GP), enabling principled initialization by maximizing the marginal likelihood over hyperparameters governing neural activation functions. Finally, we introduce an active learning strategy to efficiently select informative datapoints, further reducing training time and data requirements [1].

## 2    Background: low-rank recurrent neural networks

Consider a population of $d$ rate-based neurons with membrane potentials $\mathbf{x} = [x_1, \ldots, x_d]^\top$ and firing rates $\phi(\mathbf{x}) = [\phi(x_1), \ldots, \phi(x_d)]^\top$, where $\phi(\cdot)$ is a scalar nonlinearity mapping the membrane potential to firing rate (e.g., sigmoid, tanh, ReLU). The dynamics of a generic RNN are given by a vector ordinary differential equation and a linear output:

$$\dot{\mathbf{x}} = -\mathbf{x} + J\phi(\mathbf{x}) + B\mathbf{u}, \quad \mathbf{z} = W\phi(\mathbf{x}) \tag{1}$$

where $J \in \mathbb{R}^{d \times d}$ is the recurrent weight matrix, $B \in \mathbb{R}^{d \times d_{in}}$ the input matrix, $\mathbf{u}$ is a $d_{in}$-dimensional input signal, and $W \in \mathbb{R}^{d_{out} \times d}$ a readout matrix. This network becomes a *low-rank RNN* if the recurrent weight matrix $J$ has reduced rank $r < d$, thus factorizing as:

$$J = MN^\top = \sum_{i=1}^{r} \mathbf{m}_i \mathbf{n}_i^\top, \quad \text{where } \{M, N \in \mathbb{R}^{d \times r}\} \tag{2}$$

Here $\mathbf{m}_i, \mathbf{n}_i$ are columns of $M, N$ respectively. In this case, the network state $x(t)$ evolves within a subspace of at most $r + d_{in}$ dimensions [3, 6], with activity in the remaining dimensions decaying due to the term $(-\mathbf{x})$. Thus, the network state vector can be expressed as:

$$\mathbf{x}(t) = M\boldsymbol{\kappa}(t) + B\mathbf{v}(t), \tag{3}$$

where $\boldsymbol{\kappa}(t)$ represents latent recurrent activity and $\mathbf{v}(t)$ denotes low-pass filtered inputs [6, 17]. Finally, $\dot{\boldsymbol{\kappa}} = F(\boldsymbol{\kappa}, \mathbf{u})$ represents the differential equation that governs the low-dimensional recurrent dynamics, where $F$ is a nonlinear function of the latent state $\boldsymbol{\kappa}$ and input $\mathbf{u}$.

---

[1]Code: https://github.com/anushri10/Efficient-Training-of-Minimal-and-Maximal-Low-Rank-RNNs.git

## 3 An alternate view of low-rank RNNs

Standard approaches to training low-rank RNNs involves optimizing the parameters $\{N, M, B, W\}$ via back-propagation [17]. Here we consider an alternative approach, which amounts to solving a least squares regression problem with a set of random nonlinear basis functions.

We begin by considering the problem of embedding an arbitrary low-dimensional dynamical system into a low-rank RNN. Specifically, we wish to set the model parameters so that $\mathbf{z}$ obeys the dynamics of an particular "target" ODE:

$$\dot{\mathbf{z}} = g(\mathbf{z}) \tag{4}$$

for some function $g$. We will then identify this output with the latent vector defining the network's activity in the recurrent subspace: $\mathbf{z}(t) \triangleq \boldsymbol{\kappa}(t)$. This implies that the dimensionality of the output is equal to the rank of the network, $r = d_{out}$, and constrains the output weights to be the projection operator onto the column space of $M$, that is, $W = M(M^\top M)^{-1}$. (If the target output is lower dimensional than the rank of the network $r$, we can truncate $\mathbf{z}$ to take only its first $d_{out}$ elements).

We are then left with the problem of setting the network weights $M$, $N$, and input weights $B$ so that the latent vector $\mathbf{z}(t)$ evolves according to (eq. 4). For simplicity, consider the rank-1 case where $\mathbf{z}$ is scalar. This corresponds to an RNN weight matrix $J = \mathbf{mn}^\top$. Assume that the input is also scalar, and that the input vector $\mathbf{b} \in \mathbb{R}^d$ is orthogonal to $\mathbf{m}$ (although we relax this constraint in SI C). The network state can then be decomposed as a time-varying linear combination of $\mathbf{m}$ and $\mathbf{b}$ [16, 6, 17] (eq. 3):

$$\mathbf{x}(t) = \mathbf{m}\mathbf{z}(t) + \mathbf{b}\mathbf{v}(t), \tag{5}$$

where $\mathbf{v}(t)$ represents the low-pass filtered input, resulting from the linear dynamical system $\dot{\mathbf{v}} = -\mathbf{v} + \mathbf{u}(t)$. The fact that $\mathbf{m}$ and $\mathbf{b}$ are orthogonal means that we can write the dynamics that govern the latent variable explicitly as:

$$\dot{\mathbf{z}} = -\mathbf{z} + \mathbf{n}^\top \phi(\mathbf{m}\mathbf{z} + \mathbf{b}\mathbf{v}) \tag{6}$$

a result shown previously in [17], and which is schematized in Fig. 1. Our goal of embedding an arbitrary ODE $g(\mathbf{z})$ into the network can be now viewed as setting the model parameters so that

$$g(\mathbf{z}) + \mathbf{z} \approx \mathbf{n}^\top \phi(\mathbf{m}\mathbf{z} + \mathbf{b}\mathbf{v}) \tag{7}$$

To achieve this, note that the right-hand-side can be viewed as a linear combination of terms $\phi(m_i \mathbf{z} + b_i \mathbf{v})$ with weights $n_i$, for $i \in \{1, \ldots, d\}$. Each of these terms can be viewed as a nonlinear basis function in $\mathbf{z}$. If $\phi$ is the hyperbolic tangent function, each such term is a shifted, scaled $\tanh$ function in $\mathbf{z}$, where $m_i$ is the slope and $b_i v$ is the offset. This means that we can view the problem of embedding $g(\mathbf{z})$ into a low-rank RNN as the problem of setting $\mathbf{m}$ and $\mathbf{b}$ to build an appropriate set of basis functions, and setting $\mathbf{n}$ so that the linear combination of basis functions approximates $g(\mathbf{z}) + \mathbf{z}$. This approach formalizes the connection between low-rank RNNs and the NEF [18, 21], and shows that a low-rank RNN corresponds to a neural ODE with a single hidden layer [35–37].

Already, this perspective makes an important limitation clear: if the inputs $\mathbf{v}(t)$ are zero, the basis functions are all odd-symmetric (that is, $g(m_i \mathbf{z}) = -g(-m_i \mathbf{z})$ for all $\mathbf{z}$), crossing the origin only at zero. (see Fig. 1C). Because $-\mathbf{z}$ is also odd-symmetric, and the linear combination of odd-symmetric functions is odd-symmetric, this means that in the absence of inputs, the network can only capture the odd-symmetric component of $g(\mathbf{z})$. A low-rank RNN is therefore not a universal approximator unless it has inputs, or equivalently, different biases or offsets to each neuron (similar to general RNNs). If the $\phi$ is instead taken to be ReLU, the problem is even more severe: each basis function is a linear function with non-zero slope on either $\mathbf{z} > 0$ or $\mathbf{z} < 0$. Thus the network can only approximate $g(\mathbf{z})$ that are piecewise linear functions broken at the origin. (SI Fig. 7).

If we set the filtered input to be the constant $\mathbf{v} = 1$, we see that the problem of embedding an arbitrary ODE in a low-rank RNN amounts to fitting the ODE in a basis of shifted and scaled basis functions in $\mathbf{z}$. To achieve this, we propose to sample the scales (elements of $\mathbf{m}$) and offsets (elements of $\mathbf{b}$) to obtain a random basis, and then fit $\mathbf{n}$ by least-squares regression, namely:

$$\hat{\mathbf{n}} = (\phi(\mathbf{z}_{grid}\mathbf{m}^\top + \mathbf{b}^\top)^\top \phi(\mathbf{z}_{grid}\mathbf{m}^\top + \mathbf{b}^\top))^{-1} \phi(\mathbf{z}_{grid}\mathbf{m}^\top + \mathbf{b}^\top)^\top (g(\mathbf{z}_{grid}) + \mathbf{z}_{grid}) \tag{8}$$

where $\mathbf{z}_{grid}$ denotes a grid of points at which we wish to fit $g(\mathbf{z})$. Note that we could use weighted least squares if we care more about accurately approximating certain regions of $g(\mathbf{z})$, or add a small

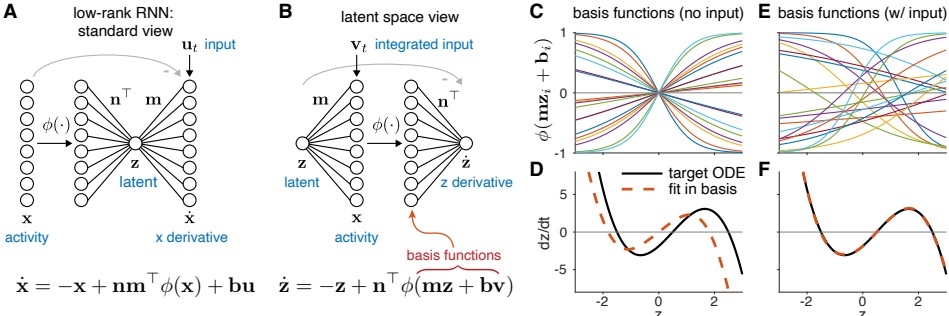

Figure 1: Two equivalent views of low-rank RNNs. **(A)** Standard view of rank-1 RNN with 8 neurons $\mathbf{x}$ and 1 latent dimension $\mathbf{z}$. **(B)** Alternate view of the same network, now framed in terms of the dynamics of latent $\mathbf{z}$. This shows that a low-rank RNN is equivalent to a neural ODE with a single hidden layer [35]. **(C)** Basis functions obtained by sampling slope parameters $m_i \sim \mathcal{N}(0,1)$, but without input ($\mathbf{v}_t = 0$). **(D)** Attempting to fit an example ODE using this basis recovers only the odd-symmetric component, since all basis functions are odd symmetric. **(E)** Adding inputs allows basis functions have random horizontal offsets. Here we sampled the input weights $\mathbf{b}_i \sim \mathcal{N}(0,1)$ and set input $\mathbf{v}_t = 1$. (Note that this could also be obtained by using per-neuron "biases"). **(F)** Least squares fitting of $\mathbf{n}$ using the basis from (E) provides good fit to the target ODE.

ridge penalty if the design matrix (whose columns are given by the basis functions evaluated at $\mathbf{z}_{grid}$) is ill-conditioned.

Fig. 1 shows an illustration of this approach for an example ODE, here chosen to be a cubic polynomial with two stable fixed points and one unstable fixed point. Note that the network cannot approximate $g(\mathbf{z})$ when the inputs are set to zero (Fig. 1C-D), but can do so with near-perfect accuracy when both the $\mathbf{m}$ vector and the (constant) inputs $\mathbf{b}$ are drawn from a Gaussian distribution (Fig. 1E-F).

### 3.1 Multi-dimensional dynamical systems

We can apply this same regression-based approach to higher-dimensional nonlinear dynamical systems, where rank $r = \dim(\mathbf{z}) > 1$. In two dimensions, the basis functions are given by $\phi(m_{1i}z_1 + m_{2i}z_2 + b_i)$, which are scaled, shifted $\tanh$ functions with a random orientation (Fig. 2A). Approximating a 2D dynamical system with a rank-2 RNN can then be written as the problem of fitting two different nonlinear functions $g_1(\mathbf{z})$ and $g_2(\mathbf{z})$ using two different linear combinations of the same 2D basis functions:

$$g(\mathbf{z}) = \begin{bmatrix} g_1(\mathbf{z}) \\ g_2(\mathbf{z}) \end{bmatrix} \approx -\begin{bmatrix} z_1 \\ z_2 \end{bmatrix} + \begin{bmatrix} \mathbf{n}_1^\top \phi(M\mathbf{z} + \mathbf{b}) \\ \mathbf{n}_2^\top \phi(M\mathbf{z} + \mathbf{b}) \end{bmatrix}, \tag{9}$$

where $M = [\mathbf{m}_1 \mathbf{m}_2]$ is a $d \times 2$ matrix whose columns define the slope and orientation of each basis function, $\mathbf{b}$ is once again a column vector of offsets, and we have assumed constant input ($\mathbf{v} = 1$). Note once again that if we do not include inputs, the basis functions are all radially odd-symmetric around the origin. Thus, once again, the RNN will only be able to capture radially odd-symmetric $g(\mathbf{z})$, and is not a universal approximator unless we include nonzero offsets $\mathbf{bv} \neq 0$.

To embed a given multi-dimensional ODE $g(\mathbf{z})$ into a low-rank RNN, we once again generate a random basis by sampling the elements of $M \in \mathbb{R}^{d \times 2}$ and $\mathbf{b} \in \mathbb{R}^d$ from a Gaussian distribution. The problem factorizes into learning each column vector $\mathbf{n}_i$ for each dimension of the $g$, we have:

$$\hat{\mathbf{n}}_i = (\phi(Z_{grid}M^\top + \mathbf{b}^\top)^\top \phi(Z_{grid}M^\top + \mathbf{b}^\top))^{-1} \phi(Z_{grid}M^\top + \mathbf{b}^\top)^\top \left(g(Z_{grid}) + Z_{grid}\right), \tag{10}$$

for $i = 1, 2$. This differs from the 1D case above only in that $Z_{grid}$ is now a $r$-column matrix of grid points, where each row contains the coordinates of a single point in $\mathbf{z}$. Note that these grid points need not be uniformly sampled; we could sample them from an arbitrary distribution, or use a collection of points from simulating the ODE from a variety of starting points (SI E.2).

Fig. 2 shows an application to an example 2-dimensional nonlinear ODE, in this case containing a stable limit cycle. Note that this 2D system is highly nonlinear and not radially odd-symmetric, so once again, embedding the system in a low-rank RNN fails if we do not include inputs (or per-neuron biases, SI A.2).

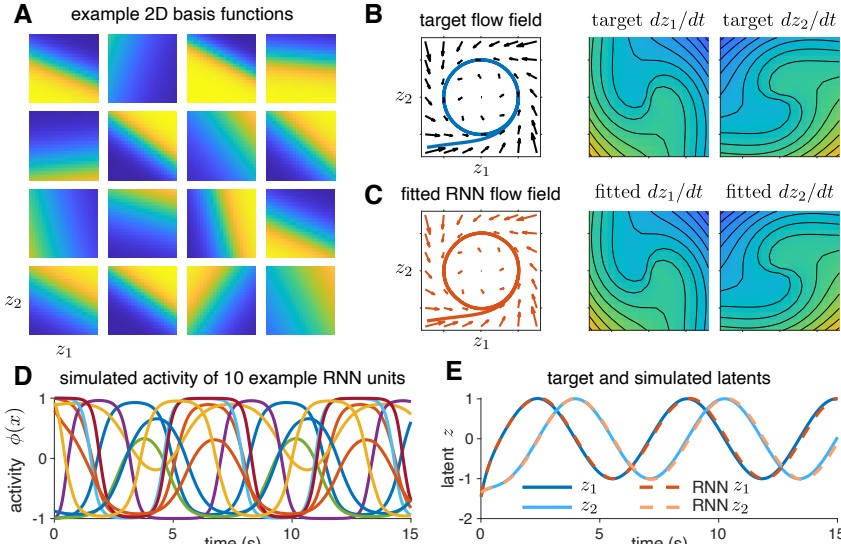

Figure 2: Embedding a 2-dimensional nonlinear ODE into a rank-2 RNN. **(A)** Example basis functions obtained by sampling $M$ and $\mathbf{b}$ coefficients from a zero-mean Gaussian, producing randomly oriented, scaled, and shifted hyperbolic tangent functions. **(B)** A target two-dimensional nonlinear dynamical system, containing a stable limit cycle on a circle of radius one, represented as a flow field (left), or by its component functions $g_1(\mathbf{z}) = \frac{dz_1}{dt}$ and $g_2(\mathbf{z}) = \frac{dz_2}{dt}$ (right). **(C)** Least squares fitting of weight vectors $\mathbf{n}_1$ and $\mathbf{n}_2$ produces a near perfect match to the target flow field, and functions $g_1$ and $g_2$. **(D)** Output firing rates $\phi(x_i)$ for 10 example units (i.e $i \in \{1, \ldots, 10\}$) during the red example trajectory shown in panel C. **(E)** Simulated trajectories from the true ODE (blue trace in panel B) and latent variable of the fitted RNN (red trace from panel C), plotted as a function of time, showing good agreement between the target ODE and the RNN output. Note that fitting was closed-form, and did not require backprop-through-time.

## 4 Comparison with backpropagation (BP) and FORCE

**Recursive least squares for online fitting**: The previous sections present an offline learning approach for $\mathbf{n}$ via a basis design matrix composed of all points on the grid ($\mathbf{z}$) at which the target ODE ($g(\mathbf{z})$) is evaluated. Here, we adapt this framework to online learning—a format used by most state-of-the-art RNN training algorithms. In online learning, at each timestep $t \in [1, 2, \cdots T]$, the network generates an output $\hat{\mathbf{z}}_t^i$ which should match the target trajectory state $\mathbf{z}_t{}^i$. Given $M$ target trajectories: $\mathbf{z}_T^i, i \in [1, 2, \cdots M]$, the objective is to minimize prediction error between network outputs and target states.

We note, our NEF approach can be modified for such a setting via online recursive updates to the weight vector $\mathbf{n}$. This translates to a recursive least-squares (RLS) optimization scheme (SI-Algorithm 1 summarizes our implementation).

**Comparison:** To evaluate the performance of our method in online learning settings, we compare against two widely-used training algorithms for RNNs: FORCE learning[9, 10] and BP. Each baseline is a trajectory-tracking framework, where the goal is to learn vector field dynamics from simulated low-dimensional trajectories.

We first evaluate on the classical sine wave generation task, originally used to benchmark FORCE [9]. Our rank-2 networks, with targets comprising both the sine signal and its cumulative integral, consistently outperform FORCE-trained full-rank networks (weights were drawn from $\mathcal{N}(0, \frac{1}{d})$) across all tested network sizes (Fig. 3. **A**). Notably, our approach achieves lower MSE while requiring fewer neurons than FORCE, which needs larger networks to suppress chaotic activity to accurately reproduce oscillatory dynamics (for additional comparisons with FORCE see SI E.3).

We next evaluate on a binary decision-making task modeled by a bistable attractor ODE. Both our method and networks trained with BP, learn from teacher trajectories that start at random initial

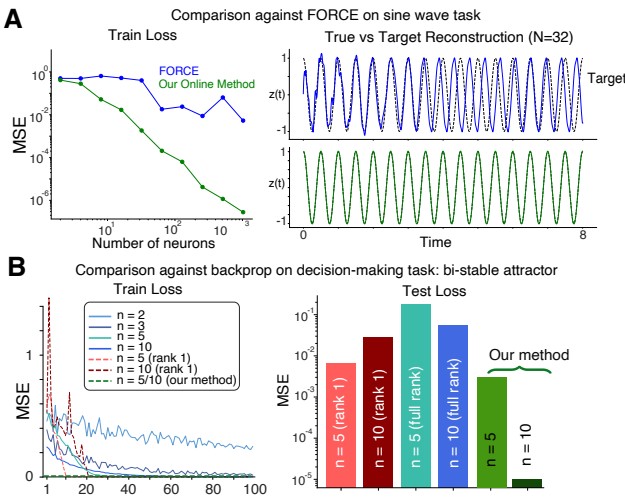

Figure 3: **(A)** Training error as a function of network size for RNNs trained via FORCE and our method on the sine-wave task. Right panel shows qualitative reconstruction for a network with 32 neurons. **(B)** Train and test error for RNNs trained to perform a binary decision-making task using bi-stable attractor dynamics [38, 39].

conditions and converge to one of two fixed points (training details: SI E). As shown in Figure 3.**B**, our approach achieves lower test error across comparable network sizes than both full-rank and low-rank BP-trained networks. Furthermore, our RLS-based approach converges with substantially reduced training time. These results highlight the efficiency of our framework for capturing structured low-dimensional dynamics.

## 5 Finding the smallest RNN for a given dynamical system

Thus far, we've introduced both offline and online methods for fitting low-rank RNNs to target dynamics, assuming access to a predefined set of nonlinear basis functions. This naturally leads to a fundamental question: *how should this basis set be chosen*? Equivalently, what is the minimal network size required to accurately approximate a given dynamical system? To address this, we now return to the offline setting, where the full target dynamics $g(\mathbf{z})$ are known in advance and $\mathbf{z}$ is a scalar. Our goal is to identify the smallest low-rank RNN—i.e., the minimal number of neurons $d' \ll d$, that can accurately implement the function $g(\mathbf{z})$. Formally, we pose this as a sparse approximation problem: find a minimal subset of basis functions $\Phi(\mathbf{m}, \mathbf{b})$ and corresponding weights $\mathbf{n}$ such that the network can faithfully reproduce the dynamics. Mathematically, this implies selecting the best $d'$ entries from $\Phi(\mathbf{m}, \mathbf{b})$, to create a basis:

$$\Phi_{d'}(\mathbf{m}, \mathbf{b}) = \begin{bmatrix} \phi(m_{i_1}\mathbf{z} + \mathbf{b}_{i_1}) \\ \vdots \\ \phi(m_{i_{d'}}\mathbf{z} + \mathbf{b}_{i_{d'}}) \end{bmatrix}, \quad \text{where } \{i_1, \ldots, i_{d'}\} \subseteq \{1, 2, \ldots, d\}.$$

Then, a linear weighting $[n_0 \ \mathbf{n}']$ is learned using least squares regression, where:

$$g(\mathbf{z}) \approx -n_0 * z + \mathbf{n}'^{\top} \Phi_{d'}(\mathbf{m}, I) \quad \text{where } \{\mathbf{n}' = 1 \times d' \text{ vector}\} \tag{11}$$

To achieve the desired optimization of approximating $g(\mathbf{z})$, we begin with a large enough $\Phi(\mathbf{m}, \mathbf{b})$ obtained by sampling from a uniform grid of values for $(\mathbf{m}, \mathbf{b})$. We then follow an iterative approach, wherein at each iteration $t$, we greedily pick a basis function $i_j$ with the highest alignment to the current residual estimate of $g(\mathbf{z})$. This is done via an adaptation of the well-established orthogonal matching pursuit (OMP) framework. A more detailed description of this process is provided in SI-Algorithm 2.

It is worth noting, changing the original basis set $\Phi(\mathbf{m}, \mathbf{b})$ to $\Phi'(\mathbf{m}, \mathbf{b})$, could result in the algorithm converging to a different minima (global minima in $\Phi'(\mathbf{m}, \mathbf{b})$ could be different from global minima in $\Phi(\mathbf{m}, \mathbf{b})$). However, if the basis sets are equivalent, we observe similar performance across simulations (SI- Fig 13)).

We also introduce a continuous extension of this framework [40, 41]. Unlike standard OMP, which selects from a discrete set of basis functions, continuous-OMP refines each selected basis function by

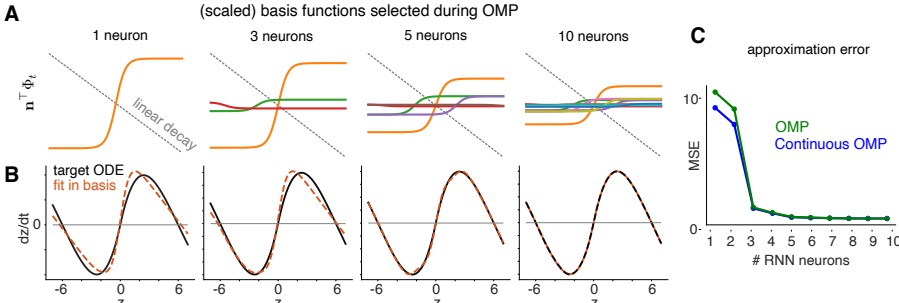

Figure 4: Finding the smallest RNN for a particular nonlinear dynamical system using orthogonal matching pursuit (OMP). **(A)** Scaled basis functions selected after $1, 3, 5$, and $10$ iterations of OMP, along with the linear decay term $-z$ for an example ODE (shown below). **(B)** Target ODE (black) and RNN fit after each step of OMP. **(C)** Mean squared error (MSE) between target ODE and RNN approximation as a function of the number of RNN neurons added by OMP, and our continuous OMP (COMP) method.

optimizing its parameters. Specifically, after each greedy selection, we optimize the corresponding $(\mathbf{m}, \mathbf{b})$ values by performing gradient descent on the ordinary least-squares objective (Eqn 11). Thus we search a continuous space of basis functions rather than being limited to a predefined discrete set, offering greater flexibility in finding the optimal representation of $g(\mathbf{z})$.

In Fig 4, we apply this method to a simulated 1D ODE, with two stable fixed points and one unstable fixed point. The first row shows the greedily-added basis functions, multiplied by their corresponding learned linear weightings. The bottom row shows their linear combination against the true underlying ODE. Through this iterative process, we observe with just $5$ neurons our network almost perfectly reconstructs the target ODE. Finally, we note our COMP method of optimization allows for fine-tuning of the basis functions' parameters, therefore leading to more accurate approximations with fewer neurons (Fig. 4C shows sharper drops in MSE for fewer neurons). We also apply this method to higher dimensional dynamics (SI D).

Critically, previous work (e.g., [39, 17]) have similar dynamics which are learned using BP with much larger networks (typically $512$ neurons). Our method instead provides an empirical framework to find the minimum number of neurons needed to fit dynamics within estimated margins of error.

## 6 Infinite low-rank RNNs

So far we have taken a "primal space" view of fitting low-rank RNNs, in which we optimize the linear weights $N$ over a fixed nonlinear basis in order to fit a dynamical system of interest. In this section we will instead rely on a dual space view in order to optimize the distribution of nonlinear basis functions employed in this representation. This view will in turn allow us to consider low-rank RNNs where the number of units goes to infinity.

For simplicity, we focus on a rank–1 network with constant filtered input ($\mathbf{v}$) given by:

$$g(z) = \sum_{i=1}^{d} \mathbf{n}_i \, \phi(\mathbf{m}_i z + \mathbf{b}_i \mathbf{v}) \tag{12}$$

Rather than fixing $d$ and optimizing the parameters $\{\mathbf{n}, \mathbf{m}, \mathbf{b}\}$ directly as we've done previously, in this section we instead take a *bayesian perspective* by placing prior distributions over the network weights. This in turn also makes $g(z)$ a random variable, through which we can then characterize the distribution over functions expressible by the network. More precisely, instead of solving the regression problem in the primal form (Eqn 8), the dual space view considers the similarity between inputs $\mathbf{z}$ and $\mathbf{z}'$ in feature space denoted by the covariance kernel depicted in Eqn 13.

$$K(z, z') = \mathbb{E}_{(\mathbf{m}, \mathbf{b})} \left[ \phi(\mathbf{m}^\top \mathbf{z} + \mathbf{b}^\top \mathbf{v}) \, \phi(\mathbf{m}^\top \mathbf{z}' + \mathbf{b}^\top \mathbf{v}) \right] \tag{13}$$

Furthermore, the Central Limit Theorem ensures that in the limit of infinite basis functions (or neurons) $d \to \infty$, the network output converges to a Gaussian Process: $g(z) \xrightarrow{d} GP(0, K)$, thus revealing that an infinitely wide low-rank RNN is *exactly* equivalent to a GP over state dynamics. Additionally, the specific form of the induced kernel is parameterized by the choice of nonlinearity $\phi$ and the distribution over $(\mathbf{m}, I)$. Specifically, when $\phi = \mathrm{erf}(\cdot)$ and the weights $(m, \mathbf{b}) \sim \mathcal{N}(0, \Sigma)$, we recover the well known arcsin analytic kernel [30, 29, 31]:

$$K_\Sigma(z, z') = \tfrac{2}{\pi} \sin^{-1}\left( \frac{2\, z^\top \Sigma z'}{\sqrt{(1 + 2z^\top \Sigma z)\,(1 + 2z'^\top \Sigma z')}} \right) \tag{14}$$

where the kernel hyperparameters $\Sigma = \{\sigma_m^2, \sigma_b^2, \sigma_{mb}\}$, characterize the distribution of this basis.

Thus, given noisy observations of the form $y_i = g(z_i) + \epsilon$, with $\epsilon \sim \mathcal{N}(0, \sigma_n^2)$, we perform GP regression and maximize the log marginal likelihood:

$$\log p(\mathbf{y} \mid \mathbf{z}, \Sigma) = -\tfrac{1}{2}\mathbf{y}^\top (K_\Sigma + \sigma_n^2 I)^{-1}\mathbf{y} - \tfrac{1}{2}\log \left\| K_\Sigma + \sigma_n^2 I \right\| - \tfrac{N}{2}\log 2\pi.$$

Optimizing this objective over $\Sigma$ selects a basis distribution that is statistically aligned with the data. Figure 5 compares GP regression fits under two such choices. Panel **A** shows the result using a standard normal prior over $(\mathbf{m}, \mathbf{b})$, as is common in the literature [3, 16, 6, 17]; the fit is poor and uncertainty remains high, especially away from the training points. Panel **B** shows the result after optimizing $\Sigma$: the posterior mean better matches the target function, and uncertainty is markedly reduced. This improvement stems from a data-adaptive kernel matrix that more faithfully captures the structure of the underlying dynamics. Finally, we train *finite rank*-1 *networks* using backprop-through-time (BPTT) and our online method (RLS) on trajectories generated from the ODE depicted in Panels A and B. We demonstrate that in both cases, GP-optimized bases yield lower training MSE than standard-normal bases (Panel **C**). These findings represent averages over 5 seeds of initialization; for additional training details see SI E.2.

Together, these results underscore the sensitivity of low-rank models to their initialization and show how a principled, inference-based approach can mitigate this sensitivity without relying on extensive trial and error.

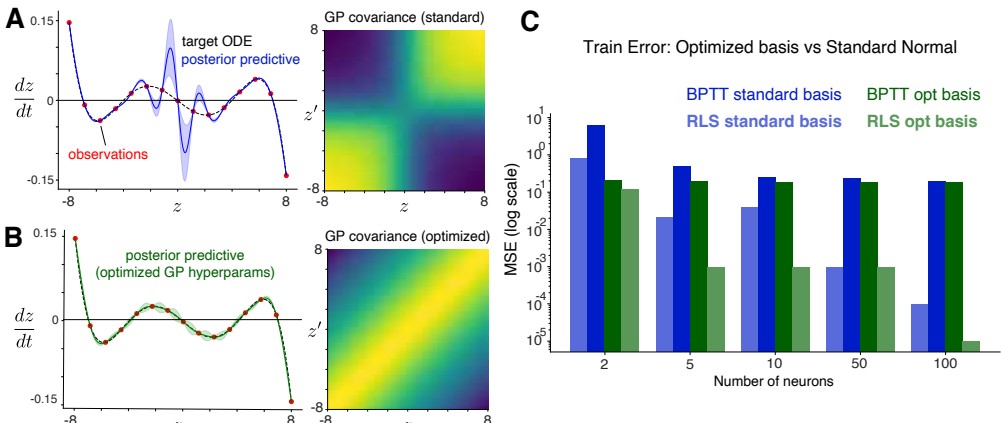

Figure 5: **Low-rank RNN as a Gaussian process.** (**A**) Targed ODE and posterior predictive distribution for an infinite RNN given a set of observations (red small). Here the GP covariance function (right) assumes a standard normal prior over basis parameters $\mathbf{m}$ and $\mathbf{b}$. (**B**) Infinite RNN predictive distribution with optimized GP covariance hyperparameters (values obtained: $\sigma_m^2 = 0.6, \sigma_b^2 = 4.5, \sigma_{mb} = 0.01$), and resulting GP covariance (right). (**C**) Comparison of networks trained by initialization from optimal basis distribution vs the standard normal distribution. Rank 1 networks trained with BPTT & our online method (RLS) achieve lower train MSE. Standard basis implies network parameters were initialized via standard normal distributions. Opt basis corresponds to intializing network weights with the GP found hyper-parameters.

# 7 Active learning for low-rank RNNs

Another consequence of the dual space view is that it naturally suggests an *active learning* strategy for sample-efficient fitting of low-rank RNNs. That is, if we can measure the ODE at a limited number of locations in latent space $\mathbf{z}$ and input space $\mathbf{v}$, where should we take those measurements? Here we build on previous work on adaptive experimental design that proposed selecting inputs that maximize information gain about the parameters of interest [42–45]. If we assume independent Gaussian noise in our measurements of $d\mathbf{z}/dt$, then the maximally informative location $(\mathbf{z}, \mathbf{v})$ is the one that maximizes our uncertainty about the weights $\mathbf{n}$, which in turn corresponds to the maximal eigenvector of the posterior covariance.

Given the design matrix $\Phi = \phi(\mathbf{m}\mathbf{z} + \mathbf{b}\mathbf{v})$ and target vector $\mathbf{y} = g(\mathbf{z}) + \mathbf{z}$, the posterior covariance over $\mathbf{n}$ under a Gaussian prior is proportional to:

$$\mathbf{\Sigma} = (\Phi^{\top}\Phi + \lambda I)^{-1}, \tag{15}$$

where $\lambda$ is the ratio of measurement noise variance to prior variance. For a candidate data-point $(z, v)$ the predictive distribution is Gaussian with variance:

$$\sigma^2(z, v) \;=\; \phi(\mathbf{m}z + \mathbf{b}v)^{\top}\mathbf{\Sigma}\,\phi(\mathbf{m}z + \mathbf{b}v) \tag{16}$$

The most informative point for the next trial is the maximizer of the predictive variance:

$$(z^{\star}, v^{\star}) \;=\; \underset{(z,v)\in\Phi}{\arg\max}\; \phi(\mathbf{m}z + \mathbf{b}v)^{\top}\big(\Phi^{\top}\Phi + \lambda I\big)^{-1}\phi(\mathbf{m}z + \mathbf{b}v), \tag{17}$$

which corresponds to the point in $(z, v)$ space with maximal projection onto the top eigenvector of $\mathbf{\Sigma}$. Intuitively, (17) targets regions where current uncertainty is maximal, therefore yielding the highest expected information gain per data point. Each selected point is added to the dataset, and the linear weight $\mathbf{n}$, is updated using the posterior mean given the data collected so far.

We illustrate this process in Fig. 6 for a system with non-normal dynamics. The heat map represents $\sigma^2(z, v)$, while red arrows depict the estimated vector field after each active data acquisition. Initially, uncertainty is highest in unexplored regions (yellow); as more samples are acquired, the learned dynamics converge rapidly to the true ODE. Furthermore, it is worth noting that within this setting we can also compute the *minimum number of samples* required to obtain a desired MSE; this bound depends on the dimensionality of the kernel function over the support of our system (i.e., the eigenspectrum of $\Phi^{\top}\Phi$ over a grid of $z, v \in [-1, +1]$), and the level of observational noise [46].

In conclusion, by exploiting the closed-form posterior in the dual (kernel) space, we obtain a computationally efficient and statistically grounded acquisition rule that significantly reduces the number of samples required to fit the system dynamics.

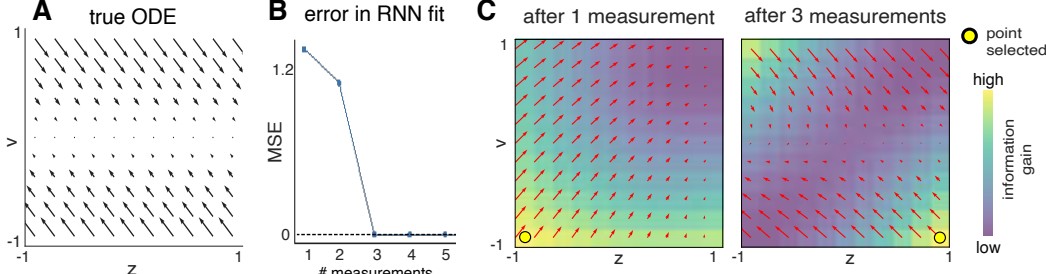

Figure 6: **Active learning for low-rank RNNs. (A)** Target ODE flow field over latent $z$ and input $v$, reflecting non-normal dynamics. **(B)** Mean squared error of rank-1 RNN approximation as a function of the number of active learning measurements. **(C)** Fitted ODE (red arrows) after 1 (left) and 3 (right) measurements selected using active learning. Heatmap shows the predictive variance (heatmap) used to compute the most informative point for the next trial (yellow dot).

# 8 Discussion

In this paper we have introduced several novel methods for low-rank RNNs, including: (1) an online learning rule for training RNNs that substantially outperforms FORCE and backprop; (2)

methods for finding minimal RNNs (i.e., RNNs with the fewest units) using an extension of OMP; (3) infinite low-rank RNNs using the equivalent Gaussian Process; and (4) active learning methods for identifying the most informative points in state space to quickly learn an approximation to an ODE of interest. Note that the third contribution provides a novel solution to the problem of optimizing the nonlinear basis functions, that is, optimizing the distribution over $M$ and $B$ parameters so that the model can accurately implement an ODE of interest. We propose maximum marginal likelihood of the equivalent GP kernel hyperparameters can thus be used to obtain improved initializations for low-rank RNNs, even in finite models that are ultimately trained with gradient based methods.

These results also open up several promising directions for future work. We have focused on the problem of embedding an ODE of interest; an important related problem is to infer low-rank networks underlying high-dimensional neural observations, a problem recently considered in [37]. Second, our active learning method assumed the ability to sample arbitrary locations in latent space, which is unrealistic in practice. A promising future direction is to develop control theoretic methods to find maximally informative inputs to the system [44]. Finally, we examined GPs equivalent to infinite RNNs for specific choices of nonlinearity (namely $\mathrm{erf}$, which we explored here, or ReLU [47]), but designing kernels that best approximate the data and then extending these kernel functions to closed-form analytic solutions remains an open problem [48].

Altogether, our contributions establish a theoretical and practical foundation for designing interpretable and sample-efficient low-rank RNNs, with broad applicability in both machine learning and neuroscience.

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

# A    Choice of nonlinearity

If we consider a network with a rectified-linear instead of a `tanh` nonlinearity, the restrictions on the network's representational capacity in the absence of inputs are even more severe (Fig. 7). In this case, the basis functions are all scaled and axis-flipped *relu* functions that intersect the $x$ axis at $x = 0$. Thus they can only represent piecewise linear functions composed of two pieces with a knot at zero. Adding inputs (or per-neuron biases) allows the network to have universal approximation capabilities.

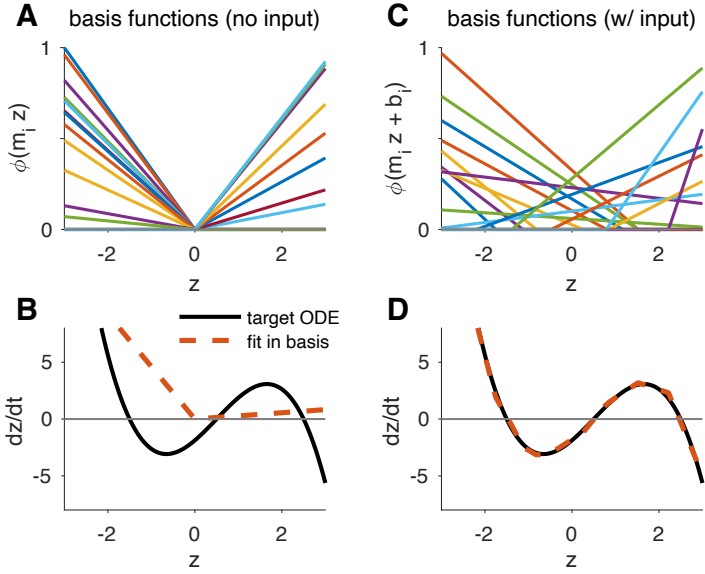

Figure 7: Representational capacity of a 1D low-rank RNN with rectified-linear (*relu*) nonlinearity. **(A)** Set of basis functions obtained by taking random coeffients $m_i \sim \mathcal{N}(0, 1)$ but without input ($\mathbf{v}_t = 0$). **(B)** Attempting to fit an example ODE using this basis recovers only a piecewise linear fit with a kink at zero. **(C)** By adding inputs, basis functions have random offset as well as slope. Here we set $\mathbf{v}_t = 1$ and sampled the input vector coefficients $\mathbf{b}_i \sim \mathcal{N}(0, 1)$. **(D)** Least squares fitting of $\mathbf{n}$ in the random basis from (C) provides a high-accuracy approximation to the target ODE.

## A.1    Comparison of activation functions in estimating ODEs

In this section, we explore the low-rank RNN's ability to approximate different types of dynamics (i.e function classes), with different activation functions (i.e basis functions). Our discussion above highlights how relu units can approximate functions through piecewise linear components. Non-zero inputs create basis functions which can be used to compose ODEs with "knots" at the shifted offsets. Alternatively, through our discussion in Section 3, we note tanh units provide smooth non-linear basis functions. The non-zero inputs create shifted basis functions, which perform a similar role, with smooth compositions. Following this intuition, if an ODE consists of smooth non-linear components it can be hypothesized that tanh units would have higher performance. Whereas, if the ODE consists of piecewise linear dynamics, relu units would prove to be more optimal. To validate this, we simulate two such ODEs in Fig. 8. Trivially, in the case of large enough number of basis functions, networks comprising of relu or tanh units can approximate any function (i.e they behave as universal approximators). However, to assess performance, we estimate the smallest networks in both cases that can fit the ODE within a pre-defined margin of error. As expected, the ODE with smoother non-linearities can be fit with smaller tanh networks than relu networks (the opposite is true for piecewise linear ODEs).

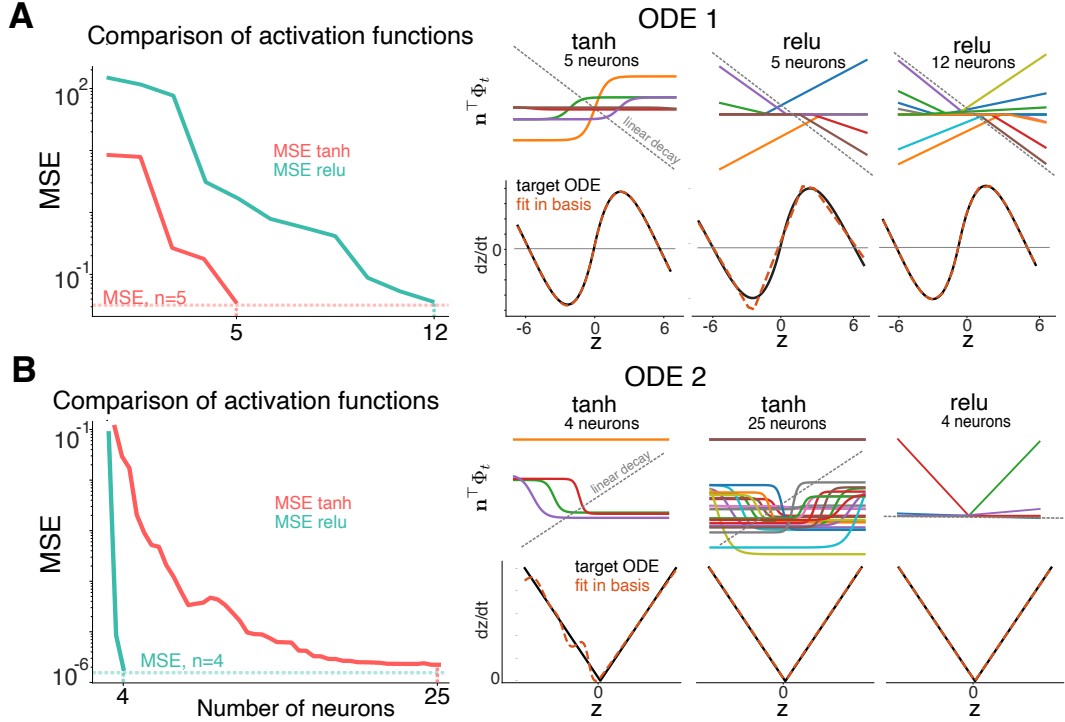

Figure 8: Performance comparison of tanh v/s relu in approximating different ODEs. **(A)** Depicts an ODE with two stable fixed points and one unstable fixed points (better fit with a tanh non-linearity). **(B)** Depicts an ODE with a shifted knot and two linear components (better fit with a RELU non-linearity). First column represents MSE (for a network with tanh and relu activations) as a function of the number of neurons in network. Neurons are added using OMP. The top row of the right column shows scaled basis functions selected via OMP. The bottom row shows fits (orange) of the target ODE (black) at the marked iterations of OMP.

## A.2 Absence of inputs for limit cycle

Section 3.1 depicts a limit cycle embedded into the low-rank RNN using our framework. The specific ODE of our non-linear and non-symmetric system is give as -

$$\frac{dx}{dt} = \left( \frac{1 - \left(z_0^2 + z_1^2\right)}{\sqrt{z_0^2 + z_1^2 + \epsilon}} \right) z_0 - z_1 - 0.35$$

$$\frac{dy}{dt} = \left( \frac{1 - \left(z_0^2 + z_1^2\right)}{\sqrt{z_0^2 + z_1^2 + \epsilon}} \right) z_1 + z_0 + 0.5$$

where $\epsilon$ is a small constant added for numerical stability. The constant values in each dimension make the underlying ODE non odd-symmetric.

In this section we show the inability of an RNN without inputs to appropriately approximate this function. In Fig. 9, the first column represents contour plots of the target ODE for each dimension. The overlayed vertical and horizontal dashed red lines depict $X = z_1 = 0, Y = z_2 = 0$ respectively. Note, there is a slight (left and upwards) shift in the contour plots, indicating the non-radial symmetry. This is introduced by adding a constant negative decay in $z_1$ and a positive correction in $z_2$. The second and third columns represents the fitted ODEs for an RNN with and without inputs respectively. It can be observed the RNN without inputs is unable to create offsets in any dimension, thus failing at recovering the underlying ODE. To further highlight this we simulate a sample trajectory from the polar coordinates of a limit cycle (detailed in Section 3.1) in the last row of Fig. 9. As expected, the low-rank RNN with inputs almost perfectly overlaps the trajectory, unlike the low-rank RNN without inputs.

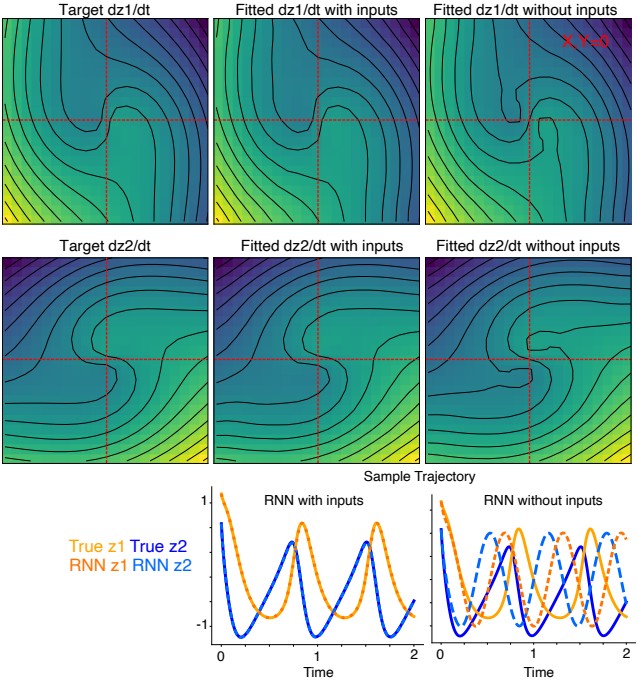

Figure 9: Influence of inputs in capturing non-symmetrical limit cycle

# B  Online recursive least square (RLS) algorithm

Here we provide additional details on our recursive least-square algorithm. Specifically, for an observed target trajectory $\{\mathbf{z}_t\}_{t=1}^T$, with the initial state $\mathbf{z}_0$:

---

**Algorithm 1** Online RLS for Low-Rank RNNs

---

1: **Inputs:** target trajectory $\{\mathbf{z}_t\}_{t=0}^T$, step size $dt$.
2: **Initialize:** Basis parameters: $\mathbf{m}$, bias $\mathbf{b}$, weights $\mathbf{n} \leftarrow \mathbf{0}$, Precision $P \leftarrow \lambda^{-1}\mathbf{I}$
3: **for** $t = 1$ **to** $T$ **do**
4:      Basis vector $\phi_t \leftarrow \phi\big(\mathbf{m}\,\mathbf{z}_t + \mathbf{b}\big)$
5:      Target derivative $y_t \leftarrow \dfrac{\mathbf{z}_t - \mathbf{z}_{t-1}}{dt}$                           (finite difference)
6:      Prediction $\hat{y}_t \leftarrow \mathbf{n}^\top \phi_t - \mathbf{z}_t$
7:      Error $e_t \leftarrow y_t - \hat{y}_t$
8:      Gain $k_t \leftarrow \dfrac{P\,\phi_t}{1 + \phi_t^\top P\,\phi_t}$
9:      Weight update $\mathbf{n} \leftarrow \mathbf{n} + k_t\,e_t$
10:     Covariance update $P \leftarrow P - k_t\,\phi_t^\top P$
11: **end for**

---

# C  General formulation & application to binary decision making task

We apply our framework to a specific group of binary decision making tasks commonly observed in systems neuroscience. In this task, a rat accumulates evidence of auditory pulses over time from clicks on its left and right side. At the end of the stimulus period, the rat must turn to the side which produced more clicks, and is rewarded for inferring this correctly. It has been shown that multiple underlying dynamical portraits could represent this behavior [39]. We thus show applicability of our method by using it to recover the autonomous and input driven dynamics on four separate synthetically generated dynamic portraits linked to this task [39]. Here, intuitively, the input dynamics encode

for the accumulation of evidence based on the clicks, and a final decision to turn is made once the accumulation value reaches a specific attractor in the network. For instance, if the instrinsic dynamics encode a bi-stable attractor, each of the end points represent a specific decision, and the inputs move the dynamics along a line between them [38]. Additionally, consistent with previous studies, we model our simulations to provide equal weights to left and right clicks but with opposite magnitudes.

We model four flow fields representing intrinsic dynamics, namely a bi-stable attractor, a line attractor, a non-canonical line attractor and the flow field inferred from [39]. More formally, they are given as follows -

Bistable attractors:
$$dz_1 = 10z_1(0.7 + z_1)(0.7 - z_1)dt + cudt$$
$$dz_2 = -10z_2dt$$

Classic DDM - line attractor:
$$dz_1 = \begin{cases} cudt & z_1 \in (-0.7, 0.7) \\ 10z_1(0.7 - z_1)(0.7 + z_1)dt & z_1 \notin (-0.7, 0.7) \end{cases}$$
$$dz_2 = -30z_2 \tag{18}$$

Non-canonical line attractor:
$$dz_1 = 5z_2$$
$$dz_2 = -5z_2dt + cudt$$

Unsupervised model:
$$dz_1 = 5z_1(0.85 + z_1)(0.85 - z_1)dt + cudt$$
$$dz_2 = 5(0.5|z_1| + 0.1)(z_1 - 1.2z_2)$$

Here, $z_1, z_2$, represent the two latent dimensions, $u$ represents the magnitude of the input clicks, and $c$ represents if its positive or negative.

Critically, we observe the input dynamics lie in a dimension *parallel* to the recurrent activity. Or alternatively, drive the system in the dimensionality spanned by the recurrent activity. We thus present a general formulation of our equations to model these input dynamics. Following Eqn 5, for a scalar $\mathbf{z}$, we now not only observe orthogonal ($\mathbf{b} = \mathbf{b}_{perp}$) neuron specific inputs, but additional input dynamics that influence the recurrent activity ($\mathbf{b}_{par}$, spans the same direction as $\mathbf{m}$), thus updating Eqn 5 as :

$$\mathbf{x}(t) = \mathbf{m}\mathbf{z}(t) + \mathbf{b}_{par}\mathbf{v}_{par}(t) + \mathbf{b}_{perp}\mathbf{v}_{perp}(t), \tag{19}$$

where $\{\mathbf{v}_{par}(t), \mathbf{v}_{perp}(t)\}$ represent the low-pass filtered inputs which drive activity along and perpendicular to the recurrent dimensions respectively.

Our goal of embedding the ODE $g(\mathbf{z})$ into the network can now be viewed as setting the model parameters so that

$$g(\mathbf{z}) + \mathbf{z} \approx \mathbf{n}^\top \phi(\mathbf{m}\mathbf{z} + \mathbf{b}_{par}\mathbf{v}_{par}(t) + \mathbf{b}_{perp}v_{perp}(t)) \tag{20}$$

This allows us to follow a similar setup to our discussions in Sec. 3, with the exception that auditory inputs are applied along $\mathbf{b}_{par}$ or $\mathbf{b} = \mathbf{b}_{perp}$, or both.

As shown in Fig 10, each row represents one of the above dynamical regimes. The first column represents the dynamics along $z_1$, or $z_2$, and the RNN fitted version. Next, we model two right (or positive) clicks at $t = 0.5$ and $t = 1$ second and a single left (negative) click at $t = 2.5$ second. The second column represents the ODE when we start from ($z = 0$), pushed by these input dynamics, for our fitted RNN dynamics (Eqn 6) against the true ODE (computed using Euler method). Lastly, we also recover the underlying flow fields, as indicated by the last column. In Fig 11, we embed a non-canonical line-attractor in which input axis is perpendicular to the line attractor and non-normal dynamics give rise to movement along the line attractor. We successfully embedded all three of these systems with rank 1 RNNs. Lastly, we also embed a system with rotational dynamics between fixed points with integration along the diagonal between them. This is done through a rank 2 RNN with inputs along each of the directions spanned by $\mathbf{b}_{par}$(Fig. 11 B). This proves the flexibility of our framework in embedding dynamics associated with neuroscience tasks.

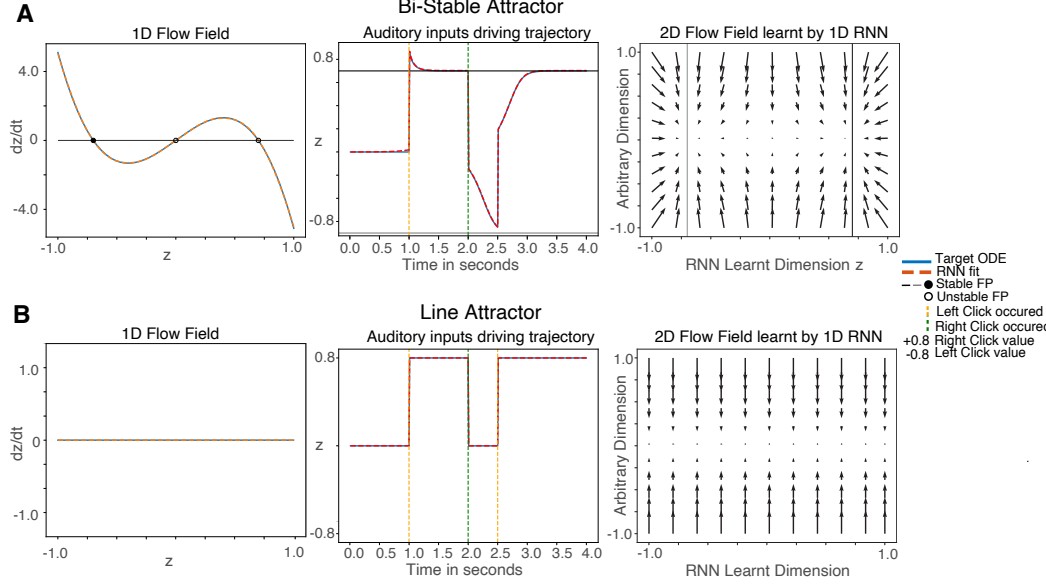

Figure 10: Two different dynamical portraits for binary decision making task: **(A)** bi-stable attractor ODE and **(B)** line attractor ODE. First column represents the true underlying ODE and the RNN estimate learned using least squares. Second column depicts a sample trajectory driven by momentary input clicks. A right click creates a drift towards the positive stable fixed point where as a left click, towards the negative stable fixed point for A. For B, accumulation along the line takes place with no diffusion. Third column represents the flow-field estimated by the RNN.

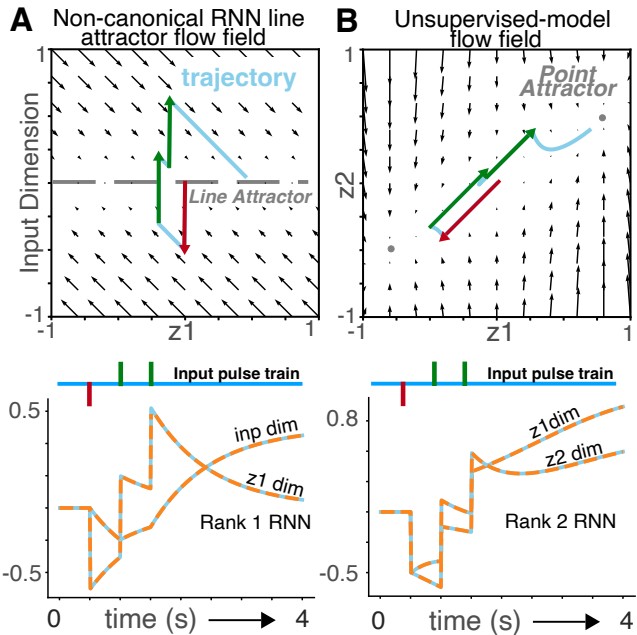

Figure 11: Two additional dynamical portraits for binary decision making task. Top: flow-field for each ODE, with input driven trajectory highlighted in blue. Bottom: true and fitted trajectories over time for each dimension.

# D OMP algorithm and it's application to a limit cycle ODE

The detailed algorithm for OMP is given here:

---

**Algorithm 2** OMP for finding smallest RNN

---

1: Select a grid of values $\mathbf{z}$.
2: Create global basis set $\Phi$ using uniformly sampled $\mathbf{m}, \mathbf{b}$ values.
3: Initialize $n$ weights using linear decay term only: $n_0 = -(\mathbf{z}^\top \mathbf{z})^{-1} \mathbf{z}^\top g(\mathbf{z})$.
4: Initialize residual: $r_0 = g(\mathbf{z}) - n_0 * (-\mathbf{z})$
5: Initialize solution basis set, $\Phi_0 = \emptyset$.
6: At each iteration $t$:

    1. Find basis vector with highest correlation with residual:

$$i_t = \arg\max_i ||\Phi_i^T \mathbf{r}_{t-1}|||  \tag{21}$$

    2. If entry $i$ is not in the solution basis:

        • Add new entry to the solution basis, $\Phi_t \leftarrow \Phi_i$
        • Solve to find new linear weights $[n_{0_t} \ \mathbf{n}'_t]$ using Eqn 8
        • Solve to find new linear weights $\mathbf{n}'_t$ using Eqn 8
        • Compute the updated residual:

$$r_t = g(\mathbf{z}) - \begin{bmatrix} n_{0_t}(-\mathbf{z}) \\ \mathbf{n}'^\top_t \Phi_t \end{bmatrix},  \tag{22}$$

    3. Check for termination based on a predefined sparsity threshold

$$d' = \text{len}(\Phi_t)  \tag{23}$$

---

## D.1 Smallest RNN for 2D Limit Cycle

Following our discussion on the multi-dimensional case and the smallest RNN (Sec. 3.1, 5 ), we apply our framework to learn the smallest number of neurons needed to fit an RNN for the limit cycle flow-field (equations provided in SI. A.2). Specifically, we apply both our OMP and COMP method to greedily add neurons that best approximate the target ODE. Fig 12 highlights the advantage of our COMP and OMP methods in designing small RNNs. Additionally we note, for our COMP method, with just 20 neurons the ODE fits are qualitatively similar to the true target, with much lower MSE values compared to OMP. This showcases the advantage of COMP as the basis parameter space increase.

## D.2 Parameter distribution of random basis for OMP

In this section we delve into the role of the distribution from which the random basis is sampled. As shown in Section 5, each basis function approximates the ODE ($g(\mathbf{z})$) over some finite domain ($\mathbf{z}$). Thus, first, it is critical the basis functions span the domain of the function being approximated. Second, these functions need not be odd symmetric and hence basis functions need to also be shifted to capture these movements. As shown in Fig 13), as long as these properties are met (i.e both the uniform grid and standard normal generate basis functions in the same domain, with the same offset ranges), the exact underlying distribution from which the basis functions are drawn does not play a critical role when finding the smallest low-rank RNN. This can be seen as the MSE values follow similar trends with greedy addition of basis functions (panel D). Qualitatively, this can also be observed via similar reconstruction of the ODE across iterations of OMP (panel A,B). Note however, by changing the distribution from which $\Phi$ is drawn the exact basis picked are different, as the global optimal basis are no longer the same (panel C).

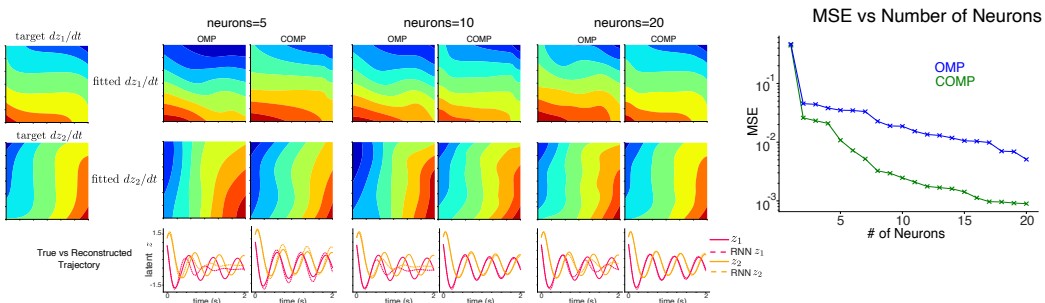

Figure 12: COMP and OMP for 2d Limit Cycle. The first row depicts the estimated flow-field for dimension one. The second row depicts the estimated flow-field for dimension two. Last row shows true vs the RNN reconstructed trajectory for both OMP and COMP. We show fits and reconstructions for $5, 10, 20$ neurons that are added via OMP and COMP respectively. Right most panel depicts the mean squared error (on log-scale) as a function of the number of neurons. Note COMP shows much steeper drops in MSE, compared to OMP.

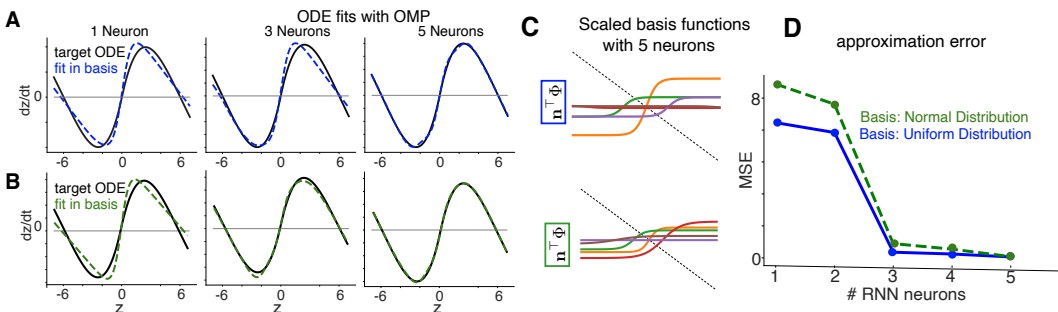

Figure 13: Influence of distribution of basis functions. **(A,B)** RNN estimated fit of Bi-stable attractor ODE the true underlying ODE over 1,3,5 iterations of OMP. In the top row basis functions (both $m_i$ and $\mathbf{b}_i$) are generated over a uniform grid spanning $+1$ to $-1$. Alternatively the bottom row consists of basis functions generated from a standard normal (i.e $m_i \sim \mathcal{N}(0,1)$, and $\mathbf{b}_i \sim \mathcal{N}(0,1)$). **(C)** Scaled basis functions selected after $5$ OMP iterations, with the top row drawn from the dniform distribution and the bottom row from the normal distribution. **(D)** Mean squared error (MSE) between target ODE and RNN approximation as a function of the number of RNN neurons added by OMP (blue: basis functions drawn from uniform grid, green: basis functions drawn from standard normal).

# E    Additional training details

## E.1    Backprop comparison for a binary decision making ODE

In Section 4, both our framework and the networks trained with Backprop are trained and tested from a set of teacher trajectories, which are simulated from the underlying ODE (eg., in the binary decision-making task, trajectories originate from random initial conditions and evolve toward one of two fixed points). To generate each trajectory, we used Euler integration with a time step of $dt = 0.01$ over a duration of $4$ seconds, yielding $400$ time steps per trajectory. A total of $160$ unique initial conditions were uniformly sampled along the $z$-axis. Of these, $150$ were used for training and the remaining $10$ for testing. Each of the networks trained via BP in Fig 3 were trained for $15$ epochs, with a batch size of $10$ per epoch. Thus, a total of $150$ gradient steps were performed (performance plateaued at $100$ gradient steps). Additionally, each of the networks were trained over three random seeds of initializations, where parameters were initialized from standard normal distributions. The values reported in Fig 3 correspond to the best performing seed. Lastly, to compare performance we trained our networks via our online RLS implementation. Once training was complete (using the $150$ training trajectories), we used the final $\mathbf{n}$ weight to test performance.

We report training times in Table 1. Note, our framework provides significantly faster training.

Table 1: Training Time For Binary-Decision Making Task

| Model Type | Size | Time(s) |
|---|---|---|
| Low Rank ($r = 1$) | 5 | 62.419 |
| Low Rank ($r = 1$) | 10 | 62.468 |
| Full Rank | 2 | 29.161 |
| Full Rank | 3 | 29.209 |
| Full Rank | 5 | 29.025 |
| Full Rank | 10 | 29.392 |
| Full Rank | 50 | 28.998 |
| Our Model | 5 | 0.069 |

## E.2 Target tracking task: Compare networks with optimal bases vs standard normal bases

To compare performance for networks trained with optimized basis against those trained with the standard normal basis, we derive a target matching problem from the ODE presented in Figure 5 (Panels A and B). Specifically, we used a set of four starting locations in $\mathbf{z}$ given by: $[7.92, 4.72, 2.3, -7.36]$. We then rolled out the dynamics (as per the ODE) using an Euler integration for a total of a 1000 time-steps, with $dt = 0.1$. Each of these trajectories (originating from each of the start locations) converged to a specific stable fixed point. We then compared rank 1 RNNs trained via our Online Method (RLS) and those trained with BPTT. Our RLS method achieves near perfect performance after a single epoch, while BPTT trained networks were trained for 5 epochs. MSE results presented are averaged over 5 random seeds of initialization.

## E.3 FORCE Comparisons

### E.3.1 Lorenz Attractor Trajectory

Following our discussion in Sec. 4, we present an additional evaluation using a more challenging target trajectory: the Lorenz attractor, a well-known chaotic system defined by the set of coupled nonlinear differential equations

$$\frac{dx}{dt} = \sigma(y - x), \quad \frac{dy}{dt} = x(\rho - z) - y, \quad \frac{dz}{dt} = xy - \beta z, \tag{24}$$

using standard parameters $\sigma = 10$, $\rho = 28$, and $\beta = 8/3$. We simulated the trajectory using Euler integration with a time step of $dt = 0.01$ for a total of $T = 3000$ steps (30 seconds of data), and used the scaled $x(t)$ signal as the target output (as in [9]). While our method was trained on (and able to reconstruct) all three variables $Z = [x(t), y(t), \text{ and } z(t)]$, only reconstructions of $x(t)$ are shown for comparison, as this is the only signal used in the FORCE training objective [9]. Specifically in our case, this can be formalized as a rank-3 RNN, written as the problem of fitting three different nonlinear functions $\frac{dx}{dt}$, $\frac{dy}{dt}$ and $\frac{dz}{dt}$ using three different linear combinations of the same 3D basis functions:

$$\begin{bmatrix} \frac{dx}{dt} \\ \frac{dy}{dt} \\ \frac{dz}{dt} \end{bmatrix} \approx - \begin{bmatrix} x \\ y \\ z \end{bmatrix} + \begin{bmatrix} \mathbf{n}_1^\top \phi(MZ + \mathbf{b}) \\ \mathbf{n}_2^\top \phi(MZ + \mathbf{b}) \\ \mathbf{n}_3^\top \phi(MZ + \mathbf{b}) \end{bmatrix}, \tag{25}$$

where $M = [\mathbf{m}_1 \mathbf{m}_2 \mathbf{m}_2]$ is a $d \times 3$ matrix, $\mathbf{b}$ is once again a column vector of offsets, and we have assumed a constant filtered input, $\mathbf{v} = 1$.

We compare networks trained using our online learning framework against those trained with the FORCE algorithm. For FORCE, we used a recurrent gain of $g = 1.5$ and trained each network across 10 random seeds; our method was similarly evaluated with the same number of seeds. Fig 14 reports mean squared error (MSE) averaged across seeds, along with representative trajectory reconstructions.

As in previous experiments, our method consistently achieves lower MSE with fewer neurons. Qualitatively, this is also evident in the trajectory reconstructions: our networks produce more

accurate fits at smaller sizes ($N = 16, 64$) than FORCE, which fails to reliably capture the signal until larger networks (for instance showed with $N = 1024$). Additionally, this further highlights that our networks converge more rapidly than those trained with FORCE.

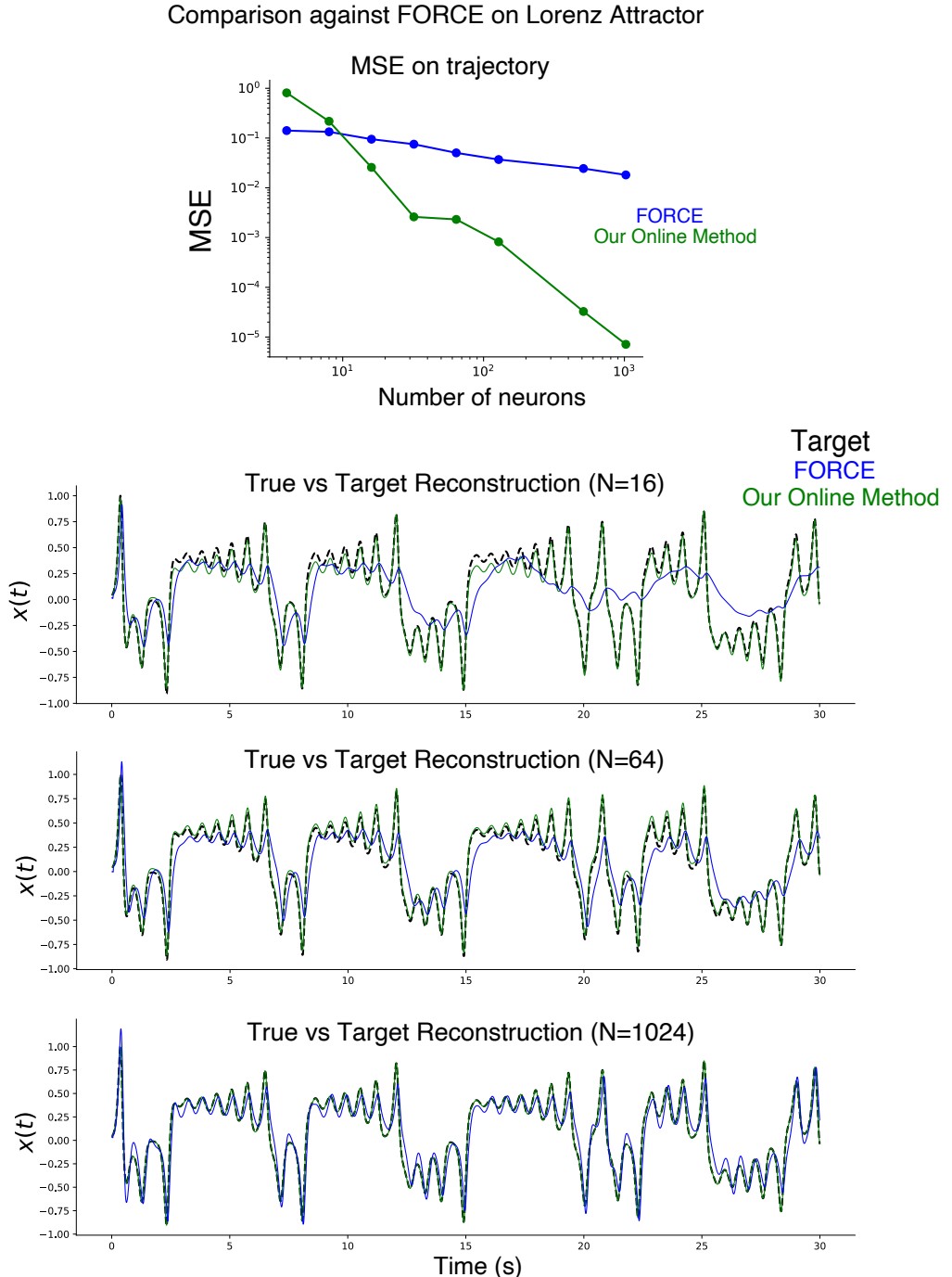

Figure 14: **Top Row:** Mean squared error (MSE) in log scale across networks of varying size (left). **Bottom Rows:** True vs reconstructed $x(t)$ trajectory from the Lorenz system, (green our method, blue FORCE) for networks of size $16, 64, 1024$.

### E.3.2 Application to noisy data

We further validate our framework by discussing it's application to noisy data. Specifically, we simulated a target signal for a total of $T = 1000$ seconds. This signal was sampled with a discrete time bin of $dt = 0.1$, thus 10,000 time steps of the target wave were obtained. We then added independent gaussian noise: $\mathcal{N} \sim (0, 0.05)$, at each time step of the trajectory to obtain the noisy target (also used in FORCE [9]). While the target output is a 1-d signal, we used the trajectory and it's two cumulative sums to fit our networks (thus needing rank 3 network). We trained RNNs using our RLS method, FORCE and BPTT (rank 3, 4). To ensure a consistent comparison, in Fig. 15 we report the training MSE (averaged across 5 seeds) after one epoch: a single pass over the 10,000-step sequence, across all methods. We see highest performance with our RLS method. Additionally, both RLS and FORCE achieve near-perfect performance in a single pass, while BPTT typically requires 5–8 epochs to reach comparable errors.

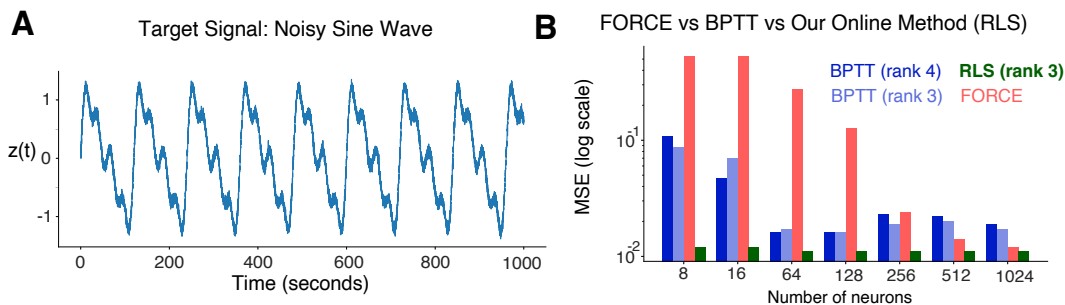

Figure 15: Networks fit on a noisy sum of four sinusoids target signal. **A**: Target trajectory sampled for $T = 1000$ seconds. **B**: Training Mean Squared Error (in log scale) of networks trained with FORCE, BPTT and our Online Method (RLS)

### E.3.3 Additional discussion on our online (RLS) method and the FORCE Framework

Below we discuss some ways in which our methodology differs from the FORCE/FULL-FORCE [9, 10] training schemes.

1. **Initialization with Full-Rank weight matrix:** FORCE/FULL-FORCE methods require full rank-initializations, and learn low-rank updates on this initilalization over time. This comes at the cost of interpretibility as the dynamics of such networks need to be analysed post training through methods such as PCA. On the other hand, our *offline and online* framework directly models the latent low-dimensional dynamic and doesn't ever need full-rank initializations.

2. **Senstive to initialization and requires multiple epochs:** One of the motivations of FORCE is that it introduces the benefits of training networks that exhibit chaotic activity prior to training. While powerful, it is observed this results in these networks needing multiple epochs/iterations. Additionally, these networks exhibit stochastic results based on initialization (e.g based on the parameter $g$). In contrast, our methods provide a deterministic closed form updates.

3. **Doesn't directly embed an ODE, but produces a set of target trajectories:** A stark difference between our *offline* framework is unlike other training methodologies similar to FORCE and FULL-FORCE that can only be trained against target trajectories, we can also directly model the underlying ODE. Thus, in cases where such a hypothesized ODE exists, we can represent the entire space of the low-dimensional dynamic.

## F  Embedding higher dimensional ODEs

To highlight the ability of our framework to embed higher dimensional ODEs, we train a network to perform the n-bit flip flop task [12]. Specifically, we consider the case where $n = 10$. In this case,

the network receives 10 binary inputs at random times, and produces 10 binary outputs. The network is expected to maintain it's output if the inputs received are the same, or alternatively switch it's output if the opposite input pulse is presented. We successfully embed such a task using a rank 10 network with 10 neurons. We design our networks such that each neuron is required to keep track of a single input pulse, thus requiring 10 neurons. Note, this could be arbitrarily scaled up to any number of input pulses, by simply scaling up the number of neurons in the network. In Fig. 16, each neuron is presented with random positive or negative input pulses in blue, to which it must respond appropriately. Our method generates the required output as seen via the activity ($x(t)$) of 4 example neurons from this network. The network was simulated for $T = 50$ seconds, sampled at $dt = 0.01$. Concretely, this was achieved by rolling out the equivalent high-dimensional network dynamics given by:

$$\dot{\mathbf{x}} = -\mathbf{x} + MN^\top \phi(\mathbf{x}) + B\mathbf{u} \tag{26}$$

In our case, $M, N, B$ are randomly drawn from standard normal distributions. $\mathbf{u}$ represents the input pulse train seen in blue, and the vector of neural activity $x(t)$ represents the desired output.

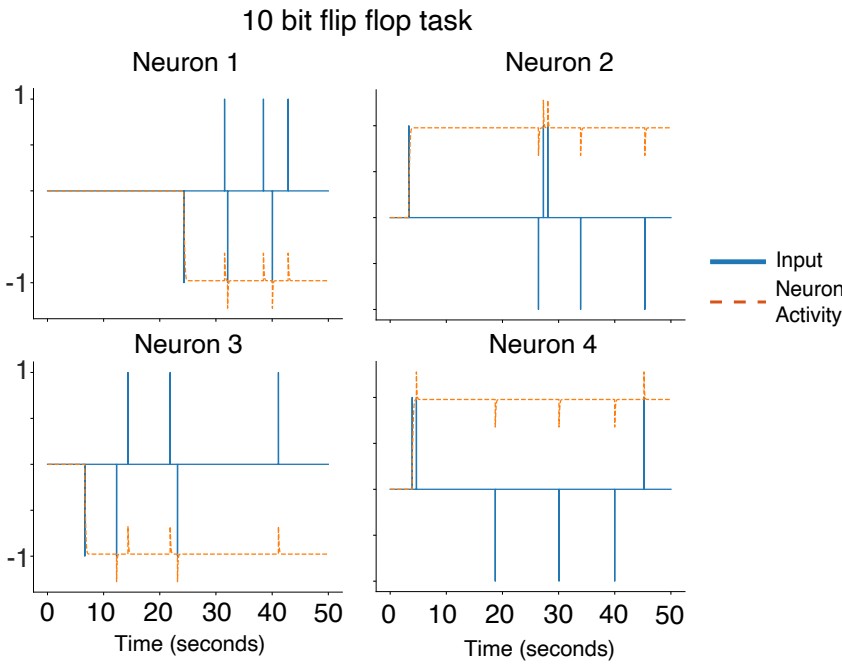

Figure 16: RNN generating appropriate response for the 10 bit flip-flop task. Neuron Activity, $x(t)$ of 4 example neurons is shown.

