# OpenReview forum: "Efficient Training of Minimal and Maximal Low-Rank Recurrent Neural Networks"
_NeurIPS.cc/2025/Conference — NeurIPS 2025 poster_

### Official Review · Reviewer_bQjY · 2025-06-07

**Clarity:** 3
**Significance:** 3
**Originality:** 3
**Rating:** 4
**Confidence:** 3

**Summary:**

This paper presents a unified framework for efficiently training low-rank recurrent neural networks (RNNs) to approximate target dynamical systems. The development of these interpretable, sample-efficient low-rank RNNs has important implications for machine learning and neuroscience applications.

**Questions:**

1. The author presents the case of v=1 in Equation (8), but the approximation of g(z) appears to perform poorly. For Iv obtained from a Gaussian distribution, how should \hat{n} be computed?

2. How to effectively approximate g(z) when the dimensionality of the target dynamics is high? In other words, is this low-rank RNN only applicable to certain low-dimensional dynamical systems?

3. This paper embeds arbitrary ODEs into an RNN network through Equation (7). What specific tasks can this RNN network be used for? If the goal is merely to analyze time-series data, why not directly apply the approximation from Equation (7) to the raw data instead of going through the RNN encoding process?

4. Other questions are detailed in the Weaknesses.

**Ethical Concerns:**

["NO or VERY MINOR ethics concerns only"]

**Final Justification:**

After reviewing the author's rebuttal, most of my concerns have been addressed. I am pleased to raise my score to 4.

**Limitations:**

yes

**Paper Formatting Concerns:**

No major formatting issues found. Paper adheres to NeurIPS 2025 guidelines.

**Quality:**

3

**Strengths And Weaknesses:**

**Strengths：**

1. Methods are well-explained with simple examples.

2. Shows advantages over standard methods (FORCE/BP) in speed and network size.


**Weaknesses:**

1. Doesn't discuss how methods perform on high-dimensional systems.

2. Only compares to older methods (FORCE/BP), not other newer low-rank RNN techniques.

---

> ### Author Rebuttal · Authors · 2025-07-30
>
> We thank the reviewer for their time & valuable feedback on our paper. We especially appreciate their positive remarks on the advantages of our method compared to FORCE/BP and the clarity of our methods.
> Below, we address specific weaknesses and questions raised by the reviewer, and humbly request the reviewer to consider raising their score if they are satisfied with our responses.
> Weaknesses:
> 1. **Performance on high dimensional systems:** The reviewer is absolutely correct in highlighting this, as indeed we have not considered high-dimensional dynamical systems. However, the primary motivation for our work is to extend the theoretical foundations of the low-rank RNN literature (e.g., Mastrogiuseppe & Ostojic, 2018; Beiran et al., 2021; Dubreuil et al., 2022; Valente et al., 2022) & the “task training” RNN literature (e.g., Mante & Sussillo 2013; Sussillo & Barak 2013; Luo, Kim, et al., bioRxiv 2023), which has focused on training RNNs to perform decision-making tasks (e.g., appendix Section C of our paper). Thus, while these tasks aren’t high-dimensional (i.e typically rank 1 or 2) , they're challenging for animals to perfect & perform. Additionally, while the ODEs considered in the main paper are either 1 or 2 dimensional, in Section G.2 of the appendix, we do show application to a 3-dimensional system (the Lorenz attractor), along with a comparison to FORCE.
>
> We would like to add, moreover, that there is no theoretical impediment to applying our framework to higher dimensional systems; the regression-based approach we have described is fully general, and thus equally applicable to systems of higher dimensionality. To illustrate this point, we will apply our method to 10, 20, and 40-dimensional systems in the revised manuscript. In fact, our marginal-likelihood based approach to fitting infinite RNNs is especially advantageous in higher dimensions, where selecting an optimal basis for a particular nonlinear function is even more critical. We feel this will substantially strengthen the paper, illustrating application to dynamical systems much higher-dimensional than those commonly considered in most neuroscience tasks.
>
> 2. **Comparison to older methods (FORCE/BP):** We thank the reviewer for highlighting this. However, to the best of our knowledge, the majority of the literature we build on (referenced in point above) uses these frameworks for training low-rank RNNs, which is why we focus our comparison on them.
>
> Questions:
> 1. **v=1, poor approximation and \hat{n} computation:** We apologize for the confusion. For v=1, our approximation is nearly perfect (please see Fig 1 panel E,F). The approximation is poor in the case of no inputs (v=0), as highlighted in Fig 1 Panel C,D. Lastly, if Iv is obtained from a Gaussian distribution, the computation remains linear in ‘n’, thus maintaining the same Least Squares solution presented in equation 8.
> 2. **Approximation of g(z) in high-dimensional settings:** Thank you for pointing this out. The approximation is achieved in the same way as the examples presented in the paper. We briefly discuss how this approximation is achieved in the following two cases:
>
>     **2.1.  If the complete ODE is known:** In the case of a high-dimensional dynamic, if the complete ODE is known, g(z)’s approximation would simply be the evaluation of the different dimensions of ‘g’. Thus each component of g(z) would be stacked to obtain the complete output. A complete description of this can be seen through our exposition of the 2-d limit cycle in Sec 3.1 (additionally Sec G.2 of the appendix extends this to the 3-d case as well).
>
>       **2.2.  Only trajectories are available (target matching scenario):** Alternatively, in the case of target matching, where trajectories from the ODE (g(z)) are available (eg see Sec 4 of the paper); we can approximate g(z) via finite differencing (Euler method). Thus $g(z) \approx \frac{z_t - z_{t-1}}{dt}$, where $dt$ represents the Euler bin.
>
> Thus the **framework presented is applicable to any dimensional ODE**, however the corresponding size of the target and design matrices would increase.
>
> 3. **Application of method to tasks and broader applicability:** Thank you for this question. The goal of our framework isn’t restricted to analyzing time-series data; rather it is designed to train a low-rank RNN to replicate a target dynamical system. Keeping this view in mind, our framework goes beyond simply approximating g(z), but rather provides an interpretable and efficient way to train low-rank RNNs.
>
>       **3.1.** Specifically as highlighted in the literature (Elia,et al.,2021; Barak 2017; Vyas, Saurabh, et al. 2020), this trained model becomes a proxy for the real system. This allows us to do experiments such as identifying minimal neurons needed, neuron ablations, perturbation analysis etc which might not be possible to carry out in the true biological or physical system (or might be costly/time intensive). Thus, such a proxy would accelerate scientific understanding of such systems. However, such models have typically been seen as black boxes, and we believe our work provides stronger theoretical and practical insights into the internal representations (and learning dynamics) of these networks.
>
>       **3.2.**  Second, we also present an active learning strategy which provides a principled way to identify informative data points, thereby reducing experimental costs.

---

> > ### Comment · Reviewer_bQjY · 2025-08-03
> >
> > Many thanks for your response. I am pleased to raise my score to 4.

---

> > > ### Author Response · Authors · 2025-08-03
> > >
> > > Thank you, we appreciate the reviewer raising their score!

---

### Official Review · Reviewer_XXEJ · 2025-06-20

**Clarity:** 4
**Significance:** 3
**Originality:** 4
**Rating:** 5
**Confidence:** 3

**Summary:**

This paper introduces an alternative perspective on low-rank RNNs. Considering the dynamics in the latent space, you can view low-rank RNNs as approximating an ODE with weighted combinations of $N$ basis functions. By randomly selecting and fixing these basis functions, you can reduce the problem of fitting a low-rank RNN to a regression problem. Online fitting basis weights with RLS converges more quickly and achieves higher performance with fewer neurons than FORCE or BPTT. This perspective also allows for finding the smallest possible RNN given a basis that approximates a given ODE with low error. The paper further shows how an infinitely-wide low-rank RNN is equivalent to a GP and optimizing the GP kernel can help find data-adaptive basis distributions. Finally, the paper show how to select data points for active learning with minimal samples.

**Questions:**

1. I assume it is still worse, but can you compare the RLS training speed to BPTT on low-rank networks while updating only $\textbf{n}$ and leaving $\textbf{m}$ and $\textbf{I}$ fixed?
2. How does the RLS-based method fare compared to FORCE and BPTT when observed trajectories have some noise?
3. Does the RNN that is fit to PCA from the bistable attractor actually qualitatively recover those dynamics?
4. Can you get similar errors with similar network sizes to OMP/COMP with just normal offline backprop (with learning the bases)? Or backprop on a larger low-rank RNN while encouraging sparsity in the rows of $\textbf{n}$ (without learning the bases)?
5. Do optimized basis distributions lead to faster training times with FORCE/BPTT/RLS for finite-size low-rank RNNs?

These last questions are probably unreasonable to address in the short rebuttal period, but I am simply sharing them anyway out of curiosity:
1. How does minimal number of neurons scale with latent dimensionality, both for good performance on RLS-based online learning and for OMP/COMP method? (Admittedly I am not sure what kind of systems you should use for these. However, I think William Gilpins's `dysts` package includes some chaotic attractors with as many as 6 dimensions.)
2. Given a low-rank RNN fit to some neural data (e.g., from Pals et al. 2024 or Valente et al. 2022), does this basis function perspective offer any kind of new insight or interpretability? Is there, for example, interesting structure in the selected basis functions?

**Ethical Concerns:**

["NO or VERY MINOR ethics concerns only"]

**Final Justification:**

The work is interesting, thorough, and well-presented. The additional text and experiments strengthen the results and should make the contributions of the work clearer to readers. However, though the authors reasonably argue for the potential value of their approach, the general applicability of this work to neuroscience remains to be seen. For this reason, I recommend a score of 5.

**Limitations:**

Yes

**Quality:**

4

**Strengths And Weaknesses:**

**Strengths:**
I found this view on low-rank RNNs to be quite interesting. It leads to several clever methods and ideas that the paper demonstrates are pretty effective, at least in simpler cases. The paper itself is also quite well-written and easy to understand. In all, I think there are a lot of exciting directions that the ideas presented here can be further developed and I am excited to see what comes of them.

**Weaknesses:**
From a neuroscience perspective, I feel that practical applications of the results shown here are somewhat limited.
* All examples in the paper are fairly idealized and very low-dimensional (mostly 1 or 2D). I imagine the number of bases required grows exponentially with latent dimensionality and that the RLS-based method might struggle when the fixed random bases are insufficient to actually approximate the system. The main examples also do not have any kind of noise from what I can tell. I would think the RLS-based method still works well, but there is no evidence of that.
* Fitting an RNN to a known latent space ODE feels to me to be a rare use case in computational neuroscience. As the authors note, RNNs are more commonly used to either fit high-dimensional neural observations or perform some (simplification of) a neuroscientific/cognitive task, and in neither case do we know what the latent dynamics should be. The example in the appendix fitting the low-rank RNN to PCA trajectories from noisy neural observations is not very compelling to me. The development of latent dynamics models was motivated in part by the noisiness of single-trial PCA, and I don't see how fitting the RNN really offers anything that the PCA did not already provide.
* Similarly, finding the minimal-size RNN that implements an ODE is also not a problem I have seen before or see much benefit in solving, and it requires full knowledge of the latent dynamics. I am happy to be convinced otherwise, though.
* Infinite low-rank RNNs, while a nice theoretical/conceptual framework, are not really practical. Though the example with optimizing the covariance is cool, I would like to see whether the benefits of this initialization transfer to fitting actual finite low-rank RNNs.
* Though the paper mentions interpretability a few times, I don't think it made a particularly strong case that it offers new methods/insights for neuroscientific interpretability. I think the perspective of approximating latent dynamics with per-neuron basis functions is nice, but I would like to see an example where, for instance, a low-rank RNN fit to neuroscientific data can be analyzed and interpreted in this fashion.

In all, I think this work is interesting and exciting, but the results and examples are quite idealized and simplified. Of course, these limitations are clearly acknowledged and extending these ideas beyond idealized settings to actual practical application is (rightfully) left to future work, but I would like to see at least some preliminary results pushing in that direction to have more confidence that these ideas are applicable to challenges neuroscientists face with noisy data and complex higher-dimensional dynamics.

---

> ### Author Rebuttal · Authors · 2025-07-30
>
> We thank the reviewer for their time and positive assessment on the originality and clarity of our work. We’re extremely grateful that they recognize the value of our contributions and share our excitement about the future directions this line of research can take!
> Below we address the specific weaknesses and questions raised by the reviewer:
>
> **Limited applicability to neuroscience**: We appreciate the reviewer for giving us a chance to address this, and apologize for our lack of discussion along these lines in the paper. We believe our work provides an alternative way to accelerate neuroscientific research. Specifically, typically such analyses of task dynamics is done in a data-driven manner. That is, task-trained RNNs interpret governing dynamics of specific tasks linked to brain computation (e.g., Kanitscheider & Fiete 2017; Turner, E, et al. 2021; Luo, Kim, et al. 2023). While this approach has numerous advantages, our lack of understanding of how such networks embed solutions leads to degeneracies in the network/task solution space therefore often leading to incorrect/evolving conclusions. It is for this reason, we propose our work as an alternative — “hypothesis-driven” analysis. To elaborate, we propose designing ODEs to solve tasks (or have access to trajectories from such ODEs), which our method then embeds into the low-rank RNN. As highlighted in our work, this reverse first process allows us to make interesting and concrete explanations regarding the network connectivity, the minimal number of neurons, ideal network initializations and a first step to closed loop experiments via active learning. We do however agree with the reviewer, that this is limited to tasks for which experimentalists/computational collaborators can effectively hypothesize such underlying low-dimensional dynamics. We will add a discussion on this in the final version of the paper.
>
> Below we address specific weaknesses highlighted by the reviewer:
> 1. **Fairly idealized, low-dimensional and noise-free examples**: The reviewer is correct in their intuition, as the latent dynamic dimension increases the required basis dimensionality also increases. While the examples in the paper are mostly 1 or 2D; we have included the **3D Lorenz attractor example and comparison of our RLS method with FORCE in the appendix (Section G.2.1). Note, in this case 16 neurons (for our RLS method) suffices for almost perfect reconstruction**.
> To the reviewers **second point on noise-free examples, we have included new analyses on a noisy target** from a sum of four sinusoids (as done in the original FORCE paper). As can be seen from our analysis, the reviewer’s intuition is correct, i.e our RLS method converges faster and with far fewer neurons ( details under response to Q2 below).
> 2. **Fitting low-rank RNNs to known ODEs/ PCA on him dim noisy data**: We thank the reviewer for highlighting this. On fitting low-rank RNNs to known ODEs – we point the reviewer to our response to the main weakness highlighted by them (Limited applicability to neuroscience). Once again, while we agree designing such an ODE for a potential task might not always be possible, we believe that in the cases that it is (such as typical 2AFC decision making/ cognitive tasks studied in this literature), our framework provides a rigorous and insightful solution.
>     - Point on PCA: we agree with the reviewer, our goal was simply to illustrate that our framework can also fit such trajectories if desired. However, we believe embedding the latent dynamics into a trained low-rank RNN offers capabilities beyond what PCA (since it’s a linear dimensionality reduction) alone provides such as understanding the influence of perturbations etc.
> 3. **Minimal size RNN:** The reviewer is correct in their assessment, the main advantage of OMP/COMP is from a computational perspective of fitting small networks.
> 4. **Infinite low-rank RNNs:** We appreciate the reviewer for highlighting this! To address this comment we’ve conducted **new analysis**, please see our result details under Q5 below.
> 5. **Fit to real data:** The reviewer’s assessment is correct. Our goal was to present interpretability from a computational perspective. I.e our method provides a geometric/algebraic description on how each neuron (in the artificial network) is a non-linear basis function of the ODE of interest. We agree there *might be* intriguing neuroscientific interpretations of this framework, however exploring these directions is left for future work.
>
> **Questions:**
> 1. **Training time RLS, BPTT (only n):** We simulated a target signal (T=1000s, dt =0.1s, thus 10,000 samples) of a sum of four sinusoids. We then added independent gaussian noise $\mathcal N(0,0.05)$ at each time step of the trajectory, thus obtaining a noisy target as seen in FORCE. While the output is a 1-d signal, we used the trajectory and two cumulative sums as our target (thus a rank 3 network). We trained low-rank networks with BPTT only allowing ‘n’s weights to be updated. Networks were trained between 5-8 epochs (i.e to achieve similar error as our method) and average running time is reported below across 5 seeds. **Note, our RLS method converges faster.**
> | Number of Neurons| Method | Time (s) |
> |----------|----------|----------|
> | 8 | RLS (rank 3)  | 0.07 |
> | 8 | BPTT  ( rank 3)  | 105.28 |
> | 8 | BPTT  ( rank 4)  | 83.29 |
> | 16 | RLS (rank 3)  | 0.12 |
> | 16 | BPTT  ( rank 3)  | 70.01 |
> | 16 | BPTT  ( rank 4)  | 59.35 |
> | 64 | RLS (rank 3)  | 0.28 |
> | 64 | BPTT  ( rank 3)  | 138.82 |
> | 64 | BPTT  ( rank 4)  | 128.69 |
> | 128 | RLS (rank 3)  | 0.64 |
> | 128 | BPTT  ( rank 3)  | 153.50 |
> | 128 | BPTT  ( rank 4)  | 28.92 |
> | 256 | RLS (rank 3)  | 1.41 |
> | 256 | BPTT  ( rank 3)  | 131.4 |
> | 256 | BPTT  ( rank 4)  | 140.03 |
> | 512 | RLS (rank 3)  | 8.34 |
> | 512 | BPTT  ( rank 3)  | 130.0 |
> | 512 | BPTT  ( rank 4)  | 130.93 |
> | 1024 | RLS (rank 3)  | 41.21 |
> | 1024 | BPTT  ( rank 3)  | 129.89 |
> | 1024 | BPTT  ( rank 4)  | 128.98 |
> 2. **RLS v FORCE v BPTT:** We use the same noisy sum of four sine-waves target discussed above. RNNs are trained with BPTT and all weights are updated. To ensure a consistent comparison, we report the training MSE (averaged across 5 seeds) after **one epoch (a single pass over the 10,000-step sequence)**. Both RLS and FORCE achieve near-perfect performance in a single pass **(note our method has highest performance)**, while BPTT typically requires 5–8 epochs to reach comparable errors. Here, we report BPTT results after one epoch to highlight the difference in time efficiency (as we can’t include images of training loss curves).
> | Number of Neurons| Method | Train Error (MSE)  | Std Error |
> |----------|----------|----------|----------|
> | 8 | RLS (rank 3)  | 0.012 |0.006 |
> | 8 | FORCE  | 0.525 | 0.064 |
> | 8 | BPTT (rank 3) | 0.107 | 0.106 |
> | 8 | BPTT (rank 4) | 0.087 | 0.0919 |
> | 16 | RLS (rank 3)  | 0.012 | 0.006 |
> | 16 | FORCE  | 0.530 | 0.029 |
> | 16 | BPTT (rank 3) | 0.047 | 0.048 |
> | 16 | BPTT (rank 4) | 0.070 | 0.0768 |
> | 64 | RLS (rank 3)  | 0.011 | 0.006 |
> | 64 | FORCE  | 0.275 | 0.056 |
> | 64 | BPTT (rank 3) | 0.016 | 0.003 |
> | 64 | BPTT (rank 4) | 0.017 | 0.006 |
> | 128 | RLS (rank 3)  | 0.011 | 0.006 |
> | 128 | FORCE  | 0.127 | 0.058 |
> | 128 | BPTT (rank 3) | 0.016 | 0.003 |
> | 128 | BPTT (rank 4) | 0.016 | 0.003 |
> | 256 | RLS (rank 3)  | 0.011 | 0.006 |
> | 256 | FORCE  | 0.024 | 0.006 |
> | 256 | BPTT (rank 3) | 0.023 | 0.011 |
> | 256 | BPTT (rank 4) | 0.019 | 0.009 |
> | 512 | RLS (rank 3)  | 0.011 | 0.006 |
> | 512 | FORCE  | 0.014 | 0.002 |
> | 512 | BPTT (rank 3) | 0.022 | 0.013 |
> | 512 | BPTT (rank 4) | 0.020 | 0.009 |
> | 1024 | RLS (rank 3)  | 0.011 | 0.006 |
> | 1024 | FORCE  | 0.012 | 0.001 |
> | 1024 | BPTT (rank 3) | 0.019 | 0.004 |
> | 1024 | BPTT (rank 4) | 0.017 | 0.003 |
> 3. **Qualitative recovery:** Yes, we achieve MSE within 10^-3, and the flow-fields when reproduced do also qualitatively match.
> 4. **Sparsity of n:** This is an interesting performance comparison which we will work on including in our final version of the paper! Thank you for bringing this to our attention.
> 5. **Optimized basis:** As per the reviewer’s suggestion we have also included **new analysis on finite sized low-rank RNNs trained with standard normal vs optimized basis** for the ODE depicted in Fig 5 of our paper. We re-frame this problem as a target matching problem. Specifically, we used a set of four starting locations given by $[7.92, 4.72, 2.3, -7.36]$. We then rolled out the dynamics (as per the ODE in Fig 5) using an Euler integration for a total of 1000 time-steps (with dt=0.1). Each of these trajectories converged to a specific stable fixed point. We then compared low-rank RNNs (rank=1) trained via our RLS method (trained for 1 epoch) and those trained with BPTT (trained for 5 epochs) on standard normal initialized basis and the optimal basis distributions found via our GP framework. Below we summarise the Train MSE. **Note, in both finite size RNNs trained with our RLS and those trained with BPTT, the optimized basis results in lower train errors.**
> | Number of Neurons| Method | Train MSE |
> |----------|----------|----------|
> | 2 | RLS  |  0.847|
> | 2 | RLS (opt)  | 0.125 |
> | 2 | BPTT  | 6.519 |
> | 2 | BPTT  (opt)  | 0.211|
> | 5 | RLS  | 0.021 |
> | 5 | RLS (opt)  | 0.001 |
> | 5 | BPTT  | 0.508 |
> | 5 | BPTT  (opt)  | 0.199|
> | 10 | RLS  | 0.041 |
> | 10 | RLS (opt)  |0.001 |
> | 10 | BPTT  | 0.250 |
> | 10 | BPTT  (opt)  |0.187 |
> | 50 | RLS  | 1e-3|
> | 50 | RLS (opt)  | 1e-3 |
> | 50 | BPTT  | 0.239 |
> | 50 | BPTT  (opt)  | 0.1907|
> | 100 | RLS  |1e-4 |
> | 100 | RLS (opt)  | 1e-5 |
> | 100 | BPTT  | 0.200 |
> | 100 | BPTT  (opt)  | 0.188|
>
> **Additional Questions:**
> We thank the reviewer for bringing these to our attention. These suggestions seem very interesting! While not in the scope of this rebuttal, we will include these analysis in future work!

---

> > ### Comment · Reviewer_XXEJ · 2025-08-01
> >
> > Thank you for the response! I think the new text and experiments are great additions to the paper, especially the optimized basis initialization result with *both* RLS and BPTT, so I will maintain my positive rating.

---

> > > ### Author Response · Authors · 2025-08-02
> > >
> > > Thank you for acknowledging the new results and for your positive assessment!

---

### Official Review · Reviewer_YAqz · 2025-06-26

**Clarity:** 3
**Significance:** 2
**Originality:** 2
**Rating:** 5
**Confidence:** 4

**Summary:**

This paper aims to design low-rank recurrent neural networks to approximate specified low-dimensional nonlinear dynamical systems, drawing on ideas from the neural engineering framework. Its main results are: 1. to show that this procedure outperforms other strategies for online training of a low-rank RNN, 2. to use an OMP-based method to find the smallest number of neurons required to approximate a given system, 3. to consider the limit of an infinitely-large network, in which their method becomes kernel regression with an NNGP-type kernel, and 4. to show that the kernel regression formulation allows for active learning.

**Questions:**

1. The formulation of Section 3 amounts to approximating the flow of interest using random feature ridge regression, and I think the paper could be enhanced by explicitly leveraging what is known about regression for random features. For instance, there is a long line of work on how many random features are required to achieve optimal error rates with respect to the number of observations, notably the work of [Rudi and Rosasco (2017)](https://proceedings.neurips.cc/paper_files/paper/2017/hash/61b1fb3f59e28c67f3925f3c79be81a1-Abstract.html). This literature has seen noticeable developments in recent years with the development of sharp asymptotic descriptions for the error assuming random sampling, see e.g. [Defilippis et al. (2024)](https://proceedings.neurips.cc/paper_files/paper/2024/hash/bd18189308a4c45c7d71ca83acf3deaa-Abstract-Conference.html) or [Atanasov et al. (2024)](https://arxiv.org/abs/2405.00592). Thus, reifying this link could shed new light on the issue of how many neurons are required to approximate a given system.

2. In Section 6, it might be interesting to comment on the fact that one can engineer kernels by clever choice of nonlinearity $\phi(\cdot)$. See [Simon et al. (2022)](https://proceedings.mlr.press/v162/simon22a.html). However, this would likely run into issues of biological plausibility.

3. The discussion of active learning is very interesting, but I wish the authors said more about the effect of sampling in $(z,v)$ in the *offline* setting. Under what assumptions on the dynamics function $g(z)$ can you say something concrete about the number of samples required? In the context of question (1), this should be possible if one fixes a probability distribution over $z$.

4. In the comparison with backpropagation through time, can you provide some details on what parameter initialization was used? I am curious if you can see a minimal-rank bias of GD from small initialization that recovers the minimum rank required to solve the task, as has been observed in various recent works from Omri Barak's group and in linear networks by [Bordelon et al. (2025)](https://arxiv.org/abs/2503.18754).

5. I wish the authors devoted more prose to the various reasons why one would want to embed a known ODE into a recurrent neural network, and what new insights this particular method could enable. For instance, do you think one could use this framework to find novel low-rank RNN implementations of commonly-studied computations? Elaborating on these broader motivations might help build a more cohesive narrative for the paper's results.

**Ethical Concerns:**

["NO or VERY MINOR ethics concerns only"]

**Final Justification:**

I had no major concerns with the initially submitted manuscript, and the authors have adequately replied to my comments and those of the other referees. I maintain my positive assessment.

**Limitations:**

yes

**Quality:**

3

**Strengths And Weaknesses:**

On the whole, I think this paper collects a set of interesting, albeit not entirely surprising, results. My main concern is that the authors do not make full use of their mapping of low-rank RNN training to a feedforward network approximation problem, as I detail below and under **Questions**. With the mapping in hand, I don't find some of their claims to be particularly novel given what is known in the feedforward network literature. For instance, the limitations on expressivity without inputs reflect the known limitations on the expressivity of feedforward networks without bias terms (in the ReLU case this is something of a folklore theorem, see e.g. [this blog post](https://james-simon.github.io/blog/the-expressivity-of-shallow-relu-nets/) or [this recent TMLR paper](https://openreview.net/forum?id=Ucpfdn66k2)). More positively, I think the authors could make use of the mapping to obtain stronger results, e.g. they could use known results about random feature models to give better estimates of how many neurons are required. With all of this in mind, I am in favor of accepting this paper, as I think the not-quite-cohesive whole into which it combines mostly existing ideas is nonetheless sufficiently interesting.

---

> ### Author Rebuttal · Authors · 2025-07-30
>
> We thank the reviewer for their time and positive assessment of our work. We especially appreciate their constructive suggestions on improving our work.
>
> Below we address specific questions raised by the reviewer:
> 1. **Optimal error rates and number of neurons:** We’re grateful for being pointed to this literature, it certainly relates to our empirical results on OMP/COMP, and active learning. We agree with the reviewer that connecting our results with the theoretical literature on optimal error rates will certainly strengthen the paper, and we will add a discussion (and citations) to this in our final version!
> 2. **Clever kernel choices:** Yes certainly! Thanks for pointing this out, we will include a reference to this.
> 3. **Effect of sampling (z,I):** Our active learning framework builds on previous results in bayesian optimal/adaptive experimental design. Specifically, we cast the problem in a linear-gaussian setting and use the Bayes A-optimal criteria (i.e maximizer of predictive posterior variance) to choose the next point. Within this setting, we can compute the *minimum number of samples* required to obtain a desired MSE; this bound depends on the dimensionality of the kernel function over the support of our system (i.e., the eigenspectrum of $\Phi(x)^T \Phi(x)$ for a grid of ‘x’ values over this support) and the level of (Gaussian) observation noise in each sample (see Chaloner & Verdinelli 1995). We will include an explicit discussion on this in our final version!
>
>     To further validate our empirical results we also performed **additional analysis comparing our active learning method to random sampling** through for the ODE presented in Fig 6. Note, **our method achieves significantly lower error rates much quicker**. Below we summarize our results (the first data point is selected randomly for our method as well for fair comparisons), MSE is computed on average over $5$ random seeds:
> | Number of data-points | Sampling Method | MSE (True ODE vs Fitted) |
> |----------|----------|----------|
> | 1 | Random | 0.814 |
> | 1 | Ours | 0.814|
> | 2 | Random | 0.820 |
> | 2 | Ours | 0.806 |
> | 3 | Random | 0.478 |
> | 3 | Ours | 0.010 |
> | 4 | Random | 0.012 |
> | 4 | Ours | 0.006 |
> | 5 | Random | 0.011 |
> | 5 | Ours | 0.006 |
> | 6 | Random | 0.008 |
> | 6 | Ours | 0.005 |
> | 7 | Random | 0.007 |
> | 7 | Ours | 0.003 |
> | 8 | Random | 0.005 |
> | 8 | Ours | 0.002 |
> | 9 | Random | 0.002 |
> | 9 | Ours | 0.0009 |
> | 10 | Random | 0.002 |
> | 10 | Ours | 0.0008 |
>
> 4. **Initial parameter initializations and minimal rank bias:** We deeply apologize for the lack of details, which we will include in the final version of the paper. For the initial comparisons in Fig 3 (for our method and BPTT), we drew $m, I$ from $\mathcal N(0,1)$. FORCE needed initializations $\mathcal N(0,\frac{1}{d})$, where $d =$ number of neurons. For the original set of experiments, the results presented averaged over $3$ seeds of initialization.
>     - **Minimal Rank Bias:** We thank the reviewer for bringing this to our attention. We’re currently working on training full-rank networks to reproduce a noisy target ((which requires a minimum of rank 3 per our framework). We will check how different initializations correlate with the effective rank of the recurrent matrix (similar to Bordelon et al. (2025)). We'll be sure to include this comparison in the final version of the paper!
>
> 5. **Discussion on embedding known ODEs:** We apologize for our lack of discussion on this topic and will include a section in our camera-ready version. Here we provide a brief explanation on this. We believe our work provides an alternative way to accelerate neuroscientific research. Specifically, typically such analyses of task dynamics is done in a data-driven manner. That is, task-trained RNNs interpret governing dynamics of specific tasks linked to brain computation (e.g., Kanitscheider & Fiete 2017; Turner, E, et al. 2021; Luo, Kim, et al. 2023). While this approach has numerous advantages, our lack of understanding of how such networks embed solutions leads to degeneracies in the network/task solution space therefore often leading to incorrect/evolving conclusions. It is for this reason, we propose our work as an alternative — “hypothesis-driven” analysis. To elaborate, we propose designing ODEs to solve tasks (or have access to trajectories from such ODEs), which our method then embeds into the low-rank RNN. As highlighted in our work, this reverse first process allows us to make interesting and concrete explanations regarding the network connectivity, the minimal number of neurons, ideal network initializations and a first step to closed loop experiments via active learning.

---

> > ### Comment · Reviewer_YAqz · 2025-07-31
> >
> > Thank you for your response! I think the authors have done a good job addressing my questions and those of the other referees, so I'm happy to maintain my positive assessment.

---

> > > ### Author Response · Authors · 2025-08-01
> > >
> > > Thank you for acknowledging our response and for your positive assessment!

---

### Official Review · Reviewer_AJnD · 2025-06-29

**Clarity:** 3
**Significance:** 4
**Originality:** 4
**Rating:** 5
**Confidence:** 5

**Summary:**

The authors propose a training algorithm, inspired on both low-rank RNNs and the Neural Engineering Framework (NEF), that allows several advances:
- it is significantly faster than backpropagation, both in theory and in practice,
- it allows to find the smallest RNNs (the ones with the least amount of neurons) able to do a given task, as supported by a sound theory,
- it can be applied in certain cases to infinitely large networks, building a link to the neural tangent kernel regime,
- it can be extended into an active learning algorithm for data-efficient learning from dynamical trajectories.

The authors also bring some theoretical insights by bridging the frameworks of low-rank RNNs and NEF into a unified framework.

**Questions:**

1) (important) Could the authors clarify exactly how their algorithm differs from NEF technically, if there are differences. I think this would help readers fit the given algorithm within their view of RNN learning algorithms, and clarify those papers contributions.
2) (minor) lines 99-100, I do not understand if the authors meant to truncate the rank instead.
3) lines128-129: I do not understand the claim hat the network can only approximate PWL functions, since PWL functions are themselves universal approximators.
4) (minor suggestion) equations (8) and (10) could be simpler to read if a simbols were assigned to the design matrices in each, but that is a matter of personal taste.
5) (important) how is the variance of m and I chosen initially and how does it affect learnability? This seems to be overlooked in the text.
6) (important) the tasks demonstrated here are always a little too simple for this venue. The absence of a truly multi-step, or higher-dimensional, or contextual tasks could raise doubts about the applicability of the method in more realistic settings. Please consider this comment in relationship with question 5) as it seems warranted that any basis will fare well in the simple tasks demonstrated, but things might not be so simple in more complex settings.
7) Could the authors clarify exactly how optimisation of the covariance in section 6 relates to the absence of details on the choice of m and I in the intial setting? In particular, it seems to me that reference [3] does insist on non-trivial and non-standard normal choices for the basis vectors, so the claim as such might be misleading, although the method proposed does very convincingly find good parameters. In particular, this relates to the limitations of a given chosen basis: can a basis made of independent and standard normal coefficients in m and I be sufficient for all tasks? How about a basis of correlated coefficients? These questions are not sufficiently addressed.

**Ethical Concerns:**

["NO or VERY MINOR ethics concerns only"]

**Limitations:**

Yes, the authors cover limitations well.

**Paper Formatting Concerns:**

No concerns.

**Quality:**

3

**Strengths And Weaknesses:**

Quality: the work is of good quality, with extensive theoretical developments that are constantly backed by sound experiments, and a sufficiently exhaustive literature review. I think most claims are supported although a few points could require more evidence (see questions).

Clarity: the paper is very clear and well-organised and can be read in a linear fashion without necessarily referring to the cited papers. There seems to be enough information to reproduce the results independently.

Significance: by bridging a number of theoretical frameworks about vanilla RNN training, this paper brings new theoretical insights to a very active area at the intersection of computational neuroscience and ML. The contributions are numerous on both the theoretical and practical sides, and most importantly they start building a bridge between the micro and statistical scales of neural networks learning theories (ie. the very few neurons-scale, and the NTK/infinite width-scale) which is one of the major gaps in NN learning theories. On the other hand, the learning algorithm itself is very similar in its implementation to the NEF algorithm, and the novel contributions on that specific aspect could be clarified. Moreover, the authors insist on their algorithms being able to fit a dynamical system, but it is not completely clear whether they think it can be used to learn a task directly, which is something the previous algorithms including FORCE, NEF, low-rank RNNs, aimed to do.

Originality: by bridging many different aspects of NN learning theory and demonstrating their links in practice, the authors demonstrate a very deep understanding of the field and a vision to unify some of its different frameworks into a sounder theoretical foundation. I don't think I have seen previous work bridging all these different frameworks.

---

> ### Author Rebuttal · Authors · 2025-07-30
>
> We thank the reviewer for their time and enthusiastic assessment on the clarity, originality, and significance of our work. We’re pleased that they found our paper clear and well-organized and appreciated both our new theoretical and practical insights. We’re especially encouraged by their recognition of our effort to bridge multiple frameworks in RNN learning theory and to provide new insights supported by sound experiments!
>
> Below we address specific questions raised:
> 1. **Difference from NEF:** We apologize for the lack of clarity on this front. Our presentation is exactly that of NEF in the case of the entire ODE being known. We would like to highlight, although the NEF presentation is indeed low-rank, we haven’t seen any explicit literature highlighting this (to the best of our knowledge). Additionally, we believe our online RLS approach is the first formulation to adapt NEF for the target matching case. While this appears straightforward, we believe that we’re the first to present this. Similarly, our other novel contributions such as finding the smallest RNN (OMP/COMP), linking the framework to the GP formulation and active learning are all novel connections to the NEF methodology. We will include a detailed discussion on this in our final version of the paper.
> 2. **lines 99-100:** We meant to highlight a case where a task/target may need more latent dimensions than the output signal. For instance, a 1-d periodic signal would need at least 2-d latents to represent oscillations. We apologize for the confusion, and will simplify our text to more clearly indicate this.
> 3. **lines 128-129:** This refers to our discussion on networks with the relu non-linearity. Here, we highlight, similar to tanh networks in the absence of per neuron inputs (or biases), the basis functions are piecewise functions about origin only, and can thus only universally express functions that are not offsetted from the origin (Fig 7 in the appendix).
> 4. **Equations 8, 10:** We thank the reviewer for bringing this to our attention, we’ll clean up our notation in the final version of the paper!
> 5. **Variance of m,I:** We deeply apologize for the lack of details, which we will include in the final version of the paper. For the initial comparisons in Fig 3 (for our method and BPTT), we drew $m, I$ from $\mathcal{N}(0, 1)$ and for FORCE we used $\mathcal{N}(0, \frac{1}{d})$, where $d=$ number of neurons.
> 6. **Tasks presented are too simple:** We agree with the reviewer that the tasks presented in the paper are fairly simple. Our goal was to present simpler toy tasks in the main paper to highlight our theoretical advancements and to develop intuition for our presented results. Below we discuss various additional/more complex tasks that we have included.
>     - **3-d example, contextual tasks:** The reviewer is absolutely correct in highlighting this, as indeed we haven’t considered high-dimensional dynamical systems. However, the primary motivation for our work is to extend the theoretical foundations of the low-rank RNN literature (e.g., Mastrogiuseppe & Ostojic, 2018; Beiran et al., 2021; Dubreuil et al., 2022; Valente et al., 2022) & the “task training” RNN literature (e.g., Mante & Sussillo 2013; Sussillo & Barak 2013; Luo, Kim, et al., bioRxiv 2023), which has focused on training RNNs to perform **decision-making tasks (e.g., appendix Section C of our paper).** Thus, while these tasks aren’t high-dimensional (i.e typically rank 1 or 2) , they're challenging for animals to perfect & perform. Additionally, while the ODEs considered in the main paper are either 1 or 2 dimensional, in **Section G.2 of the appendix, we do show application to a 3-dimensional system (the Lorenz attractor), along with a comparison to FORCE.**
> We would like to add, moreover, that there is **no theoretical impediment to applying our framework to higher dimensional systems**; the regression-based approach we have described is fully general, and thus equally applicable to systems of higher dimensionality. To illustrate this point, we will apply our method to 10, 20, and 40-dimensional systems in the revised manuscript. In fact, our marginal-likelihood based approach to fitting infinite RNNs is especially advantageous in higher dimensions, where selecting an optimal basis for a particular nonlinear function is even more critical. We feel this will substantially strengthen the paper, illustrating application to dynamical systems much higher-dimensional than those commonly considered in most neuroscience tasks.
>     - **Noisy/Complex Target:** During this rebuttal period we have also conducted **new analyses on a more complex: noisy sum of four sine-waves target signal**. Specifically: we simulated a target signal (T=1000s, dt =0.1s, thus 10,000 samples) of a sum of four sinusoids. We then added independent gaussian noise (N(0,0.05)) at each time step of the trajectory, thus obtaining a noisy target as seen in FORCE. While the output is a 1-d signal, we used the trajectory and two cumulative sums as our target (thus requiring a rank 3 network), for our RLS method. We compared our method to low-rank RNNs trained with BPTT (where all weights were updated), as well as FORCE networks (note: all weights were sampled from $N(0,\frac{1}{\sqrt d})$ as referenced by FORCE). To ensure a consistent comparison, we report the training MSE (averaged across 5 seeds) after one epoch (a single pass over the 10,000-step sequence). **Both RLS and FORCE achieve near-perfect performance (RLS has best performance) in a single pass, while BPTT typically requires 5–8 epochs to reach comparable errors.** Here, we report BPTT results after one epoch to highlight the difference in sample efficiency (as we can’t include images of the complete training loss curve).
>
> | Number of Neurons| Method | Train Error (MSE)  | Std Error |
> |----------|----------|----------|----------|
> | 8 | RLS (r=3)  | 0.012 |0.006 |
> | 8 | FORCE  | 0.525 | 0.064 |
> | 8 | BPTT (r=3) | 0.107 | 0.106 |
> | 8 | BPTT (r=4) | 0.087 | 0.0919 |
> | 16 | RLS (r=3)  | 0.012 | 0.006 |
> | 16 | FORCE  | 0.530 | 0.029 |
> | 16 | BPTT (r=3) | 0.047 | 0.048 |
> | 16 | BPTT (r=4) | 0.070 | 0.0768 |
> | 64 | RLS (r=3)  | 0.011 | 0.006 |
> | 64 | FORCE  | 0.275 | 0.056 |
> | 64 | BPTT (r=3) | 0.016 | 0.003 |
> | 64 | BPTT (r=4) | 0.017 | 0.006 |
> | 128 | RLS (r=3)  | 0.011 | 0.006 |
> | 128 | FORCE  | 0.127 | 0.058 |
> | 128 | BPTT (r=3) | 0.016 | 0.003 |
> | 128 | BPTT (r=4) | 0.016 | 0.003 |
> | 256 | RLS (r=3)  | 0.011 | 0.006 |
> | 256 | FORCE  | 0.024 | 0.006 |
> | 256 | BPTT (r=3) | 0.023 | 0.011 |
> | 256 | BPTT (r=4) | 0.019 | 0.009 |
> | 512 | RLS (r=3)  | 0.011 | 0.006 |
> | 512 | FORCE  | 0.014 | 0.002 |
> | 512 | BPTT (r=3) | 0.022 | 0.013 |
> | 512 | BPTT (r=4) | 0.020 | 0.009 |
> | 1024 | RLS (r=3)  | 0.011 | 0.006 |
> | 1024 | FORCE  | 0.012 | 0.001 |
> | 1024 | BPTT (r=3) | 0.019 | 0.004 |
> | 1024 | BPTT (r=4) | 0.017 | 0.003 |
> - **Comparison with optimized v un-optimized basis:** Finally, we have also included **new analysis** on RNNs trained with standard normal vs optimized basis. We reframe this problem as a target matching problem. Specifically, we used a set of four starting locations given by $7.92, 4.72, 2.3, -7.36$. We then rolled out the dynamics (as per the ODE in Fig 5) using an Euler integration for a total of 1000 time-steps (with dt=0.1). Each trajectory converged to a specific stable fixed point. We then compared low-rank RNNs (rank=1) trained via our RLS method and those trained with BPTT (for 5 epochs) on standard normal initialized basis and the optimal basis distributions found via our GP framework. Below we summarise the Train MSE. **Note, in both finite size RNNs trained with our RLS method and those trained with BPTT, the optimized basis results in lower train errors.**
>
> | Number of Neurons| Method | Train MSE |
> |----------|----------|----------|
> | 2 | RLS  |  0.847|
> | 2 | RLS (opt)  | 0.125 |
> | 2 | BPTT  | 6.519 |
> | 2 | BPTT  (opt)  | 0.211|
> | 5 | RLS  | 0.021 |
> | 5 | RLS (opt)  | 0.001 |
> | 5 | BPTT  | 0.508 |
> | 5 | BPTT  (opt)  | 0.199|
> | 10 | RLS  | 0.041 |
> | 10 | RLS (opt)  |0.001 |
> | 10 | BPTT  | 0.250 |
> | 10 | BPTT  (opt)  |0.187 |
> | 50 | RLS  | 1e-3|
> | 50 | RLS (opt)  | 1e-3 |
> | 50 | BPTT  | 0.239 |
> | 50 | BPTT  (opt)  | 0.1907|
> | 100 | RLS  |1e-4 |
> | 100 | RLS (opt)  | 1e-5 |
> | 100 | BPTT  | 0.200 |
> | 100 | BPTT  (opt)  | 0.188|
>
> 7. **Optimization of cov, ref [3], limitations of a chosen basis:** We appreciate the reviewer for giving us a chance to address this. The choice of $m,I$ matches the distribution which we originally used in our comparison. As per our understanding, ref [3] and follow up work (Dubreuil, et al. 2022; Valente, A. et al. 2022 ) suggests a joint normal distribution with $m,I$ (not necessarily with $0$ mean as we have used in our implementations and theoretical results).
>     - **Mean $0$ distributions:** While for this version we have worked with $0$ mean distributions (as this cleanly relates to the arcsin kernel), including the mean as a kernel hyper-parameter has been left to future work. We will clarify our claim in the paper, thanks for pointing this out!
>     - **Limitations of a choses basis:** The reviewer brings up a very interesting point on *when* correlations with $m, I$ are needed. While we agree this is an intriguing question (one which we do not answer in this work), we provide an empirical framework to optimize such a basis, to be as effective as possible in approximating an ODE. We do however agree with the reviewer and will include an explicit discussion on this in the final version of our paper!
>
> Lastly, we believe that our framework offers the **same capabilities** (with smaller, faster trained and more interpretable networks) as that of FORCE/NEF or BP (in the case of target matching tasks), as demonstrated in our results in Fig 2, appendix G.2 and our new analysis on a noisy target trajectory.

---

> > ### Comment · Reviewer_AJnD · 2025-08-02
> >
> > I thank the authors for their strong response, and I am very much looking forward to the revised manuscript and the high-dimensional experiments. The additional results on the optimized basis are particularly interesting, and I will advise the authors to think about the links with Figure 5 and how to convey the role of the basis, I think this is a very interesting aspect of the paper and one that promises future insights.
> >
> > I am happy to stand by my positive rating and happy that other reviewers seemed to agree on the quality of the presented work.

---

> > > ### Author Response · Authors · 2025-08-02
> > >
> > > Thank you for taking the time to help improve our work and we appreciate the reviewer's positive rating!

---

### Decision · Program_Chairs · 2025-09-17

**Decision:**

Accept (poster)

**Comment:**

All reviewers agree with an accept recommendation for this paper. The AC does not find any reason to overturn the consensus in the discussion, and thus recommends accept.